# Bixbyite-type $Ln_2O_3$ as promoters of metallic Ni for alkaline electrocatalytic hydrogen evolution

Hongming Sun[1,4], Zhenhua Yan[2,4], Caiying Tian[1,4], Cha Li[1,4], Xin Feng[1], Rong Huang[1], Yinghui Lan[1], Jing Chen[1], Cheng-Peng Li [1✉], Zhihong Zhang [3] & Miao Du [1,3✉]

The active-site density, intrinsic activity, and durability of Ni-based catalysts are critical to their application in industrial alkaline water electrolysis. This work develops a kind of promoters, the bixbyite-type lanthanide metal sesquioxides ($Ln_2O_3$), which can be implanted into metallic Ni by selective high-temperature reduction to achieve highly efficient Ni/$Ln_2O_3$ hybrid electrocatalysts toward hydrogen evolution reaction. The screened Ni/$Yb_2O_3$ catalyst shows the low overpotential (20.0 mV at 10 mA cm$^{-2}$), low Tafel slope (44.6 mV dec$^{-1}$), and excellent long-term durability (360 h at 500 mA cm$^{-2}$), significantly outperforming the metallic Ni and benchmark Pt/C catalysts. The remarkable hydrogen evolution activity and stability of Ni/$Yb_2O_3$ are attributed to that the $Yb_2O_3$ promoter with high oxophilicity and thermodynamic stability can greatly enlarge the active-site density, reduce the energy barrier of water dissociation, optimize the free energy of hydrogen adsorption, and avoid the oxidation corrosion of Ni.

[1] College of Chemistry, Tianjin Key Laboratory of Structure and Performance for Functional Molecules, Tianjin Normal University, 300387 Tianjin, China. [2] Key Laboratory of Advanced Energy Materials Chemistry (Ministry of Education), College of Chemistry, Nankai University, 300071 Tianjin, China. [3] College of Materials and Chemical Engineering, Zhengzhou University of Light Industry, 450001 Zhengzhou, China. [4]These authors contributed equally: Hongming Sun, Zhenhua Yan, Caiying Tian, Cha Li. ✉email: hxxylcp@tjnu.edu.cn; hxxydm@tjnu.edu.cn

Hydrogen ($H_2$) production via water electrolysis powered by solar or wind energy technologies is envisioned as an efficient strategy to meet the rising demands for renewable and clean energy resources[1,2]. As few non-noble electrocatalysts show adequate oxygen evolution reaction (OER) performance under acidic conditions, tremendous efforts have been devoted to develop the low-cost, robust and high-efficiency electrocatalysts for hydrogen evolution reaction (HER) that can be compatible with the alkaline media[3,4]. Particularly, metallic Ni with low price, high electrical conductivity, and promising alkaline HER activity has been extensively explored as the cathode material for industrial water electrolysis almost a century ago[4–7]. Nevertheless, the alkaline HER electrocatalytic activity of metallic Ni is still far from satisfactory owing to its strong hydrogen adsorption and the lack of effective water dissociation sites for sluggish Volmer step of alkaline HER[4–7]. Moreover, metallic Ni catalyst is often subjected to serious deactivation for prolonged water electrolysis, owing to the chemical corrosion by oxygen diffusion and/or strong hydrogen adsorption[8–10]. Hence, designing and exploiting efficient and durable metallic Ni-based alkaline HER catalysts, which can meet the requirement of commercial electrolyzers, is highly appealing yet challenging.

Inducing the oxophilic species into HER electrocatalysts is a valid strategy to improve their alkaline catalytic activities, which will favor the cleaving of H–OH bonds in $H_2O$ molecule and thereby facilitate the sluggish Volmer step of alkaline HER[11–13]. The transition-metal oxides and hydroxides have been widely used as the foreign oxophilic compounds, which can couple with HER catalysts to afford highly active hybrids such as $Pt/Ni(OH)_2$, $Ni/Ni(OH)_2$, $Pt/(Fe,Ni)(OH)_2$, $Co(OH)_2/MoS_2$, $Ni/NiO$, $Ru/(Fe,Ni)(OH)_2$, etc.[9,14–18]. Nevertheless, their stability at high current density is still inferior owing to the low thermodynamic stability of these oxides and hydroxides, which readily convert to metals or low-valence species under the highly reductive potential[8,19,20]. Alternatively, lanthanide oxides with high thermodynamic stability and oxophilicity are a class of promising promoters toward water dissolution. In this context, fluorite-type ceria ($CeO_2$) has shined in a variety of catalytic fields[21,22], for which the easy conversion between $Ce^{3+}$ and $Ce^{4+}$ endows it with excellent redox capability. Moreover, the specific crystal structure and reversible valence of $CeO_2$ enable the formation of oxygen vacancy. These unique properties such as high oxophilicity, multivalence, and rich oxygen vacancies ensure the formation of strong interaction between $CeO_2$ and the active component to enhance the catalytic performances[22,23]. As a result, $CeO_2$ has been extensively applied as the "performance promoter" of numerous electrocatalysts for different reactions, such as HER, hydrogen oxidation reaction (HOR), OER, oxygen reduction reaction (ORR), methanol oxidation reaction (MOR), $CO_2$ reduction reaction ($CO_2RR$), and nitrogen reduction reaction (NRR), etc[24–30]. For example, doping $CeO_2$ with Ni, Co, $Ni_2P$, $Co_4N$, CoP or NiCo etc could remarkably enhance the HER catalytic activity in alkaline media[24,31–34]. Nevertheless, their HER performances are still not comparable to the Pt-based electrocatalysts. One possible reason is the unmanageable balance between the H-/OH-binding energy and $H_2O$-dissociation energy of the catalysts. Notably, the different degrees of oxophilicity for lanthanide metal oxides provide the opportunities to screen new water-dissociation promoters with a better matching to metallic Ni, which thus may concurrently realize the low water-dissociation energy barrier and the optimized H-/OH-binding energies. In this context, the bixbyite-type lanthanide sesquioxides ($Ln_2O_3$, Ln = Sm, Eu, Gd, Dy, Ho, Er, Tm, Yb, and Lu) are often described as the fluorite-type $CeO_2$ structure with ordered oxygen vacancies[35], thereby showing potential as the performance promoters for alkaline HER, which however are unexplored thus far.

In this work, we prepare a series of graphite plate (GP) supported $Ni/Ln_2O_3$ hybrids by the selective high-temperature reduction approach. Remarkably, the $Ni/Ln_2O_3$ electrodes illustrate higher HER activity and long-term stability relative to the Ni electrode, and the enhancement effect of $Ln_2O_3$ as excellent electrocatalytic promoters is revealed. Furthermore, the screened $Ni/Yb_2O_3$ hybrid, with both the low $H_2O$-dissociation energy barrier and the optimized H-/OH-binding energy, exhibits significantly higher electrocatalytic HER activity than the well-known $Ni/CeO_2$, revealing that $Yb_2O_3$ is a preferable promoter than $CeO_2$ for HER in alkaline condition. Additionally, the strong coupled $Yb_2O_3$ with high thermodynamic stability not only prevents the agglomerate of Ni during high-temperature sintering, but also avoids the chemical corrosion of Ni during the long-term HER tests. The current finding not only opens up the applications of cubic bixbyite-type $Ln_2O_3$ as electrocatalytic promotors, but also makes $Ni/Yb_2O_3$ a promising cathode material for commercial electrolyzers.

## Results

**Preparation and characterizations of the $Ni/Ln_2O_3$ electrodes.** A selective high-temperature reduction method was developed to synthesize the $Ni/Ln_2O_3$ electrodes (see Fig. 1a). First, the $Ni(OH)_2/Ln(OH)_3$ precursor was loaded on a graphitic substrate (Supplementary Fig. 1) by a simple $NO_3^-$ reduction electrodeposition method[36]. In the electrodeposition process, $NO_3^-$ is reduced and the produced $OH^-$ leads to the synchronous generation of $Ni(OH)_2$ and $Ln(OH)_3$[34]. The ultralow solubility of $Ni(OH)_2$ and $Ln(OH)_3$ ensures their rapid and quantitative deposition on the graphitic substrate with excellent chemical homogeneity. Taking $Ni/Yb_2O_3$ as an example, after deposition of the $Ni(OH)_2/Yb(OH)_3$ precursor, the gray graphitic substrate turns green (Supplementary Figs. 2–4). During the following high temperature sintering, $Yb(OH)_3$ will decompose into $Yb_2O_3$ below 500 °C (Supplementary Fig. 5). From the viewpoint of thermodynamics ($\triangle_r G_m^\theta = \triangle_r H_m^\theta - T\triangle_r S_m^\theta$), a high temperature beyond ca. 10,000 °C is required to reduce $Yb_2O_3$ to Yb under $H_2$ atmosphere, whereas $Ni(OH)_2$ could be reduced to metallic Ni by $H_2$ at 0 °C (Supplementary Table 1). Therefore, the $Ni(OH)_2/Yb(OH)_3$ precursor can be selectively converted to $Ni/Yb_2O_3$ at 500 °C under a $H_2/Ar$ (10%) atmosphere. Due to the similarity of thermodynamic parameters for $Ln_2O_3$, homologous $Ni/Ln_2O_3$ could be obtained by this method as well. Moreover, this selective high-temperature reduction method is available to prepare various metal/metal oxide hybrids.

The X-ray diffraction (XRD) patterns (Fig. 1b) of $Ni/Ln_2O_3$ nanoparticles scraped off the graphite plate are similar. The peaks at 28.3–29.6°, 32.6–34.4°, 47.2–49.4°, and 56.0–58.7° are attributed to the (222), (400), (440), and (622) facets of the cubic bixbyite-type $Ln_2O_3$. Figure 1c presents the bixbyite structure of $Ln_2O_3$ with a face-centered cubic (fcc) unit cell of Ln centers, which are coordinated by six nearest-neighboring oxygen atoms. The bixbyite-type $Ln_2O_3$, also known as the C-type rare-earth oxide structure according to the Goldschmidt's classification, is generally considered as the defect cubic fluorite-type $CeO_2$ with ordered oxygen vacancies (Fig. 1d)[35]. Moreover, the other diffraction peaks (Fig. 1b) at 44.5°, 51.8°, and 76.4°, respectively, are assigned to the (111), (200), and (220) facets of cubic Ni (JCPDS No. 4-850, space group $Pm3m$, Supplementary Fig. 6), which reveals the formation of $Ni/Ln_2O_3$ hybrids. The scanning electron microscopy (SEM) images of $Ni/Ln_2O_3$ electrodes indicate that the $Ni/Ln_2O_3$ nanoparticles uniformly cover over the graphitic plate (Supplementary Figs. 7–15). The Ni/Ln atomic ratios obtained from energy-dispersive spectroscopy (EDS) analysis, are all ca. 90/10 in these $Ni/Ln_2O_3$ hybrids, which are similar to the initial $Ni^{2+}/Ln^{3+}$

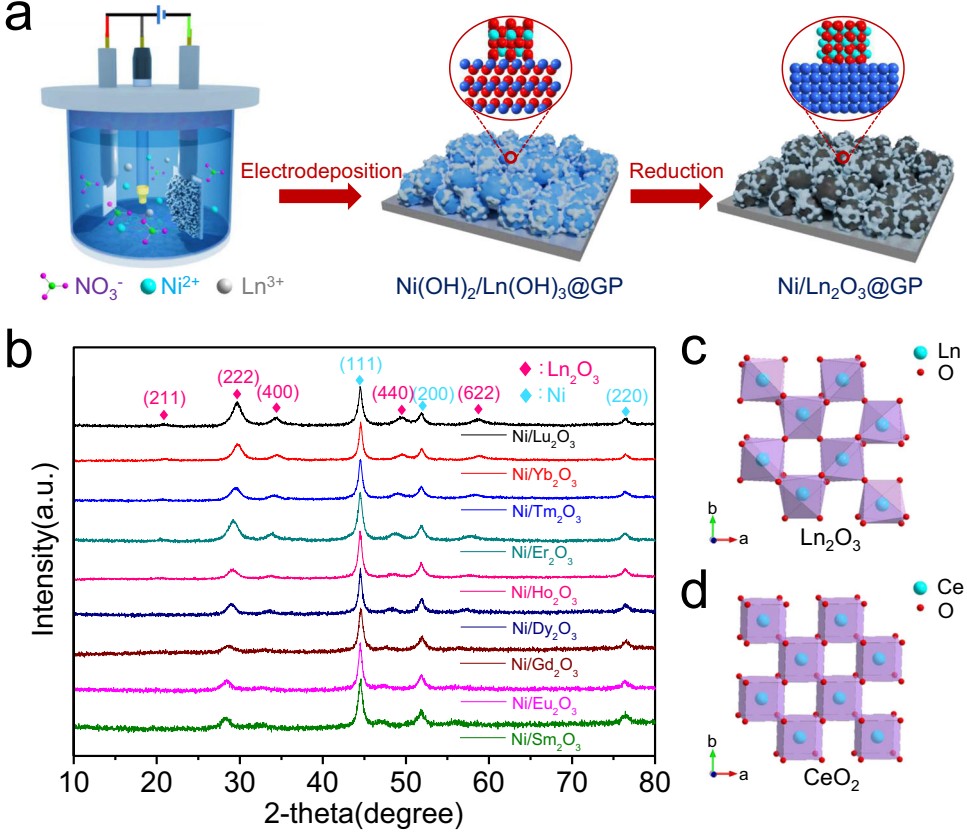

**Fig. 1 Preparation and phase analysis of electrodes. a** Synthetic scheme of graphite plate supported Ni/Ln$_2$O$_3$ electrodes. **b** XRD patterns of Ni/Ln$_2$O$_3$ hybrids. **c** Crystal structure of bixbyite-type Ln$_2$O$_3$. **d** Crystal structure of fluorite-type CeO$_2$.

feed ratio. X-ray photoelectron spectroscopy (XPS) tests of Ni/Ln$_2$O$_3$ indicate that Ni is metallic state and the Ln elements are trivalent (Supplementary Figs. 7–15). The above results confirm the successful preparation of a series of Ni/Ln$_2$O$_3$ electrodes with analogous chemical compositions, morphologies and crystal structures, thus resulting in the comparability of their HER catalytic activities. For a comparison, the pristine Ni nanoparticles on graphite plate (Ni electrode, Supplementary Fig. 16) and Ln$_2$O$_3$ nanoparticles on graphite plates (Ln$_2$O$_3$ electrodes, Supplementary Figs. 17 and 18) were prepared by the similar procedure.

**Electrocatalytic HER activities of the Ni/Ln$_2$O$_3$ electrodes.** The HER catalytic performances of Ln$_2$O$_3$ (Sm$_2$O$_3$, Eu$_2$O$_3$, Gd$_2$O$_3$, Dy$_2$O$_3$, Ho$_2$O$_3$, Er$_2$O$_3$, Tm$_2$O$_3$, Yb$_2$O$_3$, and Lu$_2$O$_3$) electrodes were tested in 1.0 M KOH (Supplementary Fig. 19). The Ln$_2$O$_3$ electrodes show negligible HER catalytic activity with high overpotentials. The linear sweep voltammetry (LSV) polarization curves (Fig. 2a) reveal that the Ni/Ln$_2$O$_3$ electrodes with Ln$_2$O$_3$ coupling show remarkably improved electrocatalytic activity than the pristine Ni electrode. Interestingly, to drive a current density of 100 mA cm$^{-2}$, the overpotentials of Ni/Ln$_2$O$_3$ electrodes decrease in turn from Ni/Sm$_2$O$_3$ (184.3 mV) to Ni/Yb$_2$O$_3$ (81.0 mV), and ultimately, slightly increase to 84.0 mV for Ni/Lu$_2$O$_3$. Remarkably, the overpotentials of Ni/Ln$_2$O$_3$ electrodes (81.0–184.3 mV) are all lower than that of Ni electrode (217.1 mV).

The Tafel slopes of electrodes were applied to judge the reaction mechanism (Fig. 2b). The lowest Tafel slope is found for Ni/Yb$_2$O$_3$ (44.6 mV dec$^{-1}$) and the highest Tafel slope is observed for Ni/Sm$_2$O$_3$ (116.7 mV dec$^{-1}$). Similarly, the Tafel slopes for Ni/Ln$_2$O$_3$ electrodes are all lower than that for Ni

electrode (124.9 mV dec$^{-1}$). The Tafel values of Ni/Ln$_2$O$_3$ electrodes reveal that the HER follows the Volmer–Heyrovsky mechanism[37,38]:

$$H_2O + e^- = H_{ads} + OH^- (\text{Volmer step}) \qquad (1)$$

$$H_2O + e^- + H_{ads} = H_2 + OH^- (\text{Heyrovsky step}) \qquad (2)$$

The high Tafel slope of Ni electrode reveals that the Volmer step is the rate-determining step[39,40]. That is, the Ln$_2$O$_3$ coupling will greatly facilitate the sluggish water dissociation process of HER on metallic Ni, and the facilitating effect increases from Sm$_2$O$_3$ to Yb$_2$O$_3$ and Lu$_2$O$_3$. A comparison of the Tafel slope and the overpotential at 100 mA cm$^{-2}$ evidently demonstrates that the Ni/Ln$_2$O$_3$ electrodes outperform the Ni electrode, and Ni/Yb$_2$O$_3$ has the highest catalytic activity among all Ni/Ln$_2$O$_3$ electrodes (Fig. 2c). The turnover frequencies (TOFs) of Ni/Ln$_2$O$_3$ and Ni electrodes were further calculated to reveal their intrinsic activities. The corresponding active sites of electrodes were quantified using electrochemical active surface areas (ECSAs) (Supplementary Figs. 20 and 21). The Ni/Ln$_2$O$_3$ electrodes perform the larger ECSAs and thus have more active sites than Ni electrode. Remarkably, the Ni/Ln$_2$O$_3$ electrodes still show higher TOF than Ni electrode after averaging over each of the active sites (Fig. 2d). Specifically, at an overpotential of 100 mV, Ni/Sm$_2$O$_3$ shows the smallest TOF (0.026 H$_2$ s$^{-1}$) and Ni/Yb$_2$O$_3$ shows the largest TOF (0.362 H$_2$ s$^{-1}$), which is 15 times higher than that of Ni electrode (0.024 H$_2$ s$^{-1}$), indicating that the Ni/Yb$_2$O$_3$ electrode has greatly enhanced intrinsic HER activity in alkaline media compared with the Ni electrode (Supplementary Fig. 22).

To understand the origin of electrocatalytic activity and the role of Ln$_2$O$_3$, electrochemical impedance spectroscopy (EIS) was

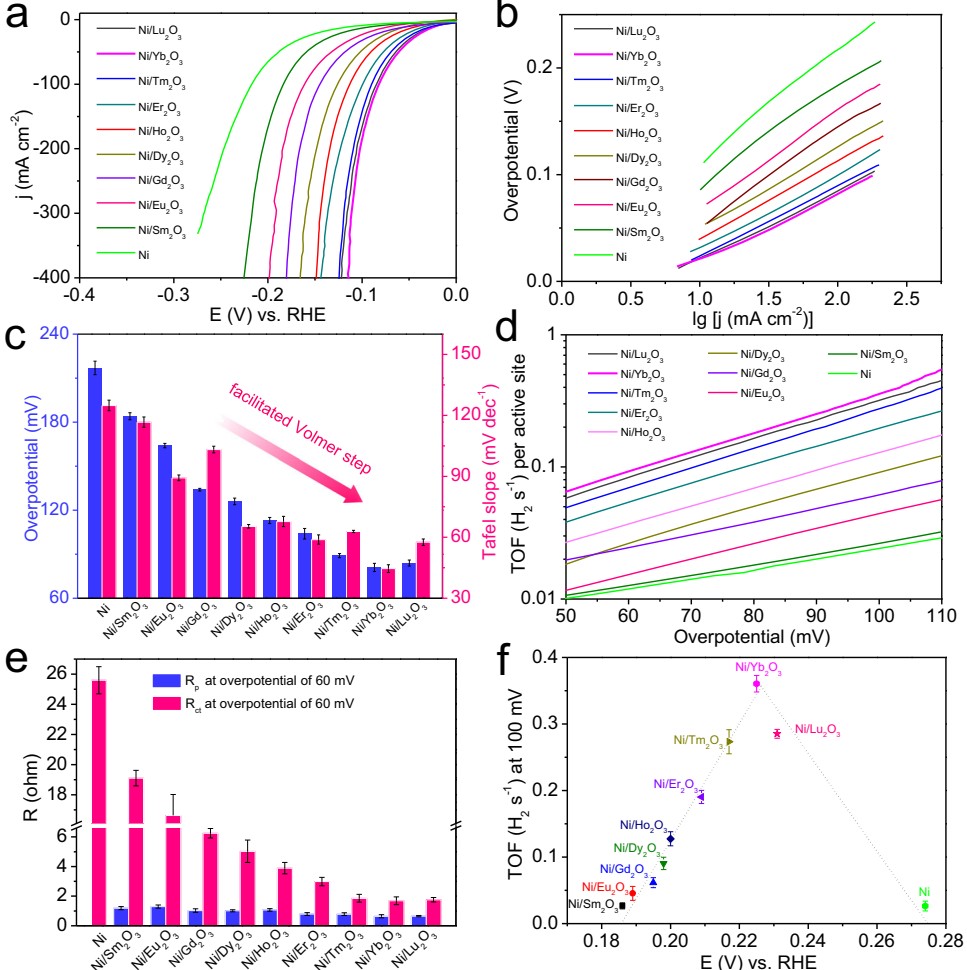

**Fig. 2 Electrocatalytic HER activity of the Ni/Ln$_2$O$_3$ electrodes in 1.0 M KOH electrolyte. a** Polarization curves (scan rate: 5 mV s$^{-1}$) of the Ni/Ln$_2$O$_3$ and Ni electrodes with mass loading of ca. 3.5 mg cm$^{-2}$. **b** The corresponding Tafel plots. **c** Comparison of catalytic activity in term of the overpotential at 100 mA cm$^{-2}$ and Tafel slopes. **d** TOF values of the Ni/Ln$_2$O$_3$ and Ni electrodes. **e** Comparison of the charge transfer resistance ($R_{ct}$) and mass transfer resistance ($R_p$) of the Ni/Ln$_2$O$_3$ and Ni electrodes. **f** Volcano plot of TOF value at 100 mV as a function of potential for the OH adsorption peak of the Ni/ Ln$_2$O$_3$ and Ni electrodes. The error bars in **c**, **e**, and **f** show the standard derivation based on triplicate measurements.

tested at different overpotentials (Supplementary Figs. 23 and 24a) for Ni/Ln$_2$O$_3$ and Ni electrodes. The Randles electrical equivalent circuit model is used to interpret the AC impedance of HER on Ni electrode without a response related to the hydrogen adsorption, and the Armstrong equivalent circuit is used to explain the AC impedance behavior on Ni/Ln$_2$O$_3$ electrodes as the second semicircle in Nyquist curves (Supplementary Fig. 24b)[41]. In Armstrong equivalent circuit model, $R_{ct}$ reveals the charge transfer resistance (low frequency semicircle) for electrode reaction, and $R_p$ indicates the mass transfer resistance (high frequency semicircle) of adsorbed intermediate $H_{ads}$[42]. The EIS spectra of Ni/Ln$_2$O$_3$ and Ni electrodes both exhibit the expected behaviors when increasing the overpotentials, that is, the total resistances decrease with the increase of overpotentials (Supplementary Figs. 23 and 24a). The main difference is found for the $R_{ct}$ values of Ni/Ln$_2$O$_3$ electrodes, which decrease from Ni/Sm$_2$O$_3$ to Ni/Yb$_2$O$_3$ and then, slightly increase for Ni/Lu$_2$O$_3$. Also, all $R_{ct}$ values of Ni/Ln$_2$O$_3$ are lower than that of Ni at the overpotentials from 20 to 100 mV (Fig. 2e, Supplementary Fig. 24c and supplementary Table 2). As the rate-determining step of alkaline HER is the Volmer reaction, the $R_{ct}$ of Ni electrode should mainly arise from the sluggish Volmer reaction[41]. Correlating the regular variations of catalytic activities with the consistent $R_{ct}$ sequences for Ni/Ln$_2$O$_3$ and Ni electrodes,

we believe that the Ln$_2$O$_3$ promotor can activate water and facilitate the sluggish water decomposition step (Volmer step) of the alkaline HER on Ni, which agrees well with the results derived from the Tafel slopes. Moreover, the $R_p$ values of Ni/Ln$_2$O$_3$ electrodes are small and not changed obviously, showing a slight decrease from Ni/Sm$_2$O$_3$ to Ni/Lu$_2$O$_3$, which indicates that the mass transfer behaviors of adsorbed intermediate ($H_{ads}$) are not dominating in regulating the catalytic activities of Ni/Ln$_2$O$_3$ electrodes.

It is known that the reaction barrier for water dissociation step of alkaline HER is governed by the adsorption energy of hydroxyl species (OH$_{ads}$)[6,43]. Herein, the incorporation of oxophilic Ln$_2$O$_3$ in Ni/Ln$_2$O$_3$ could strengthen the OH-binding energy, thereby accelerating the adsorption of water molecules and cleaving of HO–H bond[44]. To verify this inference, the cyclic voltammogram (CV) curves for OH adsorption and desorption on the surfaces of Ni/Ln$_2$O$_3$ and Ni electrodes were measured (Supplementary Fig. 25). For Ni/Ln$_2$O$_3$, an obvious negative shift of OH adsorption peak (ca. 0.186–0.231 V) compared with that of Ni (ca. 0.274 V) is observed, which shows the stronger OH-binding energy on surface of Ni/Ln$_2$O$_3$ electrodes than that of Ni electrode[45]. Moreover, the OH adsorption potentials for Ni/ Ln$_2$O$_3$ electrodes gradually decrease with a variation of decorated

$Ln_2O_3$ from $Sm_2O_3$ to $Lu_2O_3$, which is consistent with the regular changes of oxophilicity for these Ln elements in the same period. More interestingly, a volcano relation could be achieved from the TOF values at 100 mV for $Ni/Ln_2O_3$ and Ni electrodes, as a function of experimentally measured OH adsorption potentials (Fig. 2f). The volcano relation reveals that there may exist an optimal value of OH-binding energy, which reflects the Sabatier principle that the optimized electrocatalysts would adsorb the intermediates neither too strongly nor too weakly[15,46]. The Ni and $Ni/Lu_2O_3$ electrodes have the relatively weaker OH adsorption, which cannot facilitate the water dissociation effectively. In contrast, the strong OH adsorption ability from $Ni/Sm_2O_3$ to $Ni/Tm_2O_3$ can promote the water adsorption and dissociation availably, which however may also impede the OH desorption and thus block the active sites. As a result, the $Ni/Yb_2O_3$ electrode with a proper OH binding ability can realize an optimal balance between promoting the $H_2O$ dissociation and preventing the poisoning effect[43]. The regular changes of OH adsorption ability of $Ni/Ln_2O_3$ could also be confirmed by their O 1s XPS spectra collected under normal atmospheric pressure and 25 °C (Supplementary Fig. 26). The peaks of O 1s XPS spectra can be assigned to the absorbed OH groups and O atoms from $Ln_2O_3$ lattice. Obviously, the OH coverage gradually decreases from $Ni/Sm_2O_3$ to $Ni/Lu_2O_3$, which indicates the reduced adsorption strength of OH from $Ni/Sm_2O_3$ to $Ni/Lu_2O_3$[47,48].

**Analysis of microstructures and chemical environments for Ni/$Yb_2O_3$.** To obtain the deeper insights into the origin of excellent HER performances for $Ni/Yb_2O_3$, the contrast experiments and structural/component characterizations were carried out. Firstly, the $Ni/Yb_2O_3$ hybrids with diverse compositions (99:1, 97:3, 95:5, 90:10, 80:20, 70:30, and 60:40) were similarly prepared to confirm the optimal Ni:Yb molar ratio (Supplementary Fig. 27 and Supplementary Table 3). As observed in Supplementary Fig. 27, those $Ni/Yb_2O_3$—99:1, $Ni/Yb_2O_3$—97:3, $Ni/Yb_2O_3$—95:5, $Ni/Yb_2O_3$—80:20, $Ni/Yb_2O_3$—70:30, and $Ni/Yb_2O_3$—60:40 also show the similar morphology of nanoparticle arrays to that of $Ni/Yb_2O_3$ (i.e. $Ni/Yb_2O_3$—90:10). The XRD patterns (Fig. 3a) interestingly suggest that as the ratio of $Yb_2O_3$ increases, the (111) facet of cubic Ni gradually shifts to smaller diffraction angles for these $Ni/Yb_2O_3$ hybrids, which reveals the slight lattice expansion of Ni nanoparticles and the enhanced coupling between Ni and $Yb_2O_3$. Moreover, as the content of $Yb_2O_3$ increases, the crystallite size of Ni nanoparticles decreases and that of $Yb_2O_3$ increases gradually (Fig. 3b), as determined from the XRD peak widths of Ni(111) and $Yb_2O_3$(222) using the Debye–Scherrer equation. This reveals that the introduction of $Yb_2O_3$ can significantly lower the size of Ni phase, because $Yb_2O_3$ with high thermodynamic stability will prevent the agglomeration of Ni in the annealing process. The reduced sizes of Ni nanoparticles can enhance the active-site density in the $Ni/Yb_2O_3$ hybrids. As the content of $Yb_2O_3$ increases, the ECSAs of these $Ni/Yb_2O_3$ hybrids ($Ni/Yb_2O_3$—99:1, $Ni/Yb_2O_3$—97:3, $Ni/Yb_2O_3$—95:5 and $Ni/Yb_2O_3$—90:10) increases gradually. But when excessive $Yb_2O_3$ is doped, the ECSAs of the $Ni/Yb_2O_3$ hybrids will be decreased, because the smaller ECSA of $Yb_2O_3$ lowers the total ECSAs of the $Ni/Yb_2O_3$ hybrids (Supplementary Figs. 28 and 29). As a result, $Ni/Yb_2O_3$—90:10 with appropriate doping amount of $Yb_2O_3$ shows the highest ECSA among the $Ni/Yb_2O_3$ electrodes with different compositions (Supplementary Fig. 30). The result is also consistent with the higher Brunauer–Emmett–Teller (BET) specific surface area of $Ni/Yb_2O_3$ (29.0 $m^2 g^{-1}$) compared with that of Ni (18.1 $m^2 g^{-1}$), as evaluated by their $N_2$ sorption isotherms (Supplementary Fig. 31).

The transmission electron microscopy (TEM) image (Supplementary Fig. 32a) shows that $Ni/Yb_2O_3$ consists of the closely interconnected nanoparticles. The TEM mapping confirms the homogenous distribution of Ni, Yb and O elements (Fig. 3c). High-resolution TEM (HRTEM) image of $Ni/Yb_2O_3$ (Fig. 3d) indicates well-resolved lattice fringes with the interplanar spacing of 0.204 and 0.301 nm calculated from non-interface area, which could be assigned to Ni(111) and $Yb_2O_3$(222), respectively. The HRTEM images (Fig. 3d, Supplementary Fig. 32b, c) show that each Ni nanoparticle is surrounded by some $Yb_2O_3$ nanocrystallines. As indicated by TEM mapping and HRTEM images (Supplementary Figs. 33–38) of the other $Ni/Yb_2O_3$ hybrids with different Ni:Yb molar ratios ($Ni/Yb_2O_3$—99:1, $Ni/Yb_2O_3$—97:3, $Ni/Yb_2O_3$—95:5, $Ni/Yb_2O_3$—80:20, $Ni/Yb_2O_3$—70:30, and $Ni/Yb_2O_3$—60:40), the Ni and $Yb_2O_3$ nanoparticles are also mixed in form of heterojunction structures, and the $Yb_2O_3$ nanocrystallines are distributed on the surface of Ni nanoparticles. Furthermore, as the content of Yb increases, the number of $Yb_2O_3$ nanoparticles increases and the size of Ni nanoparticles decreases clearly, which is consistent with the results of particle size analysis from the XRD patterns.

In other TEM images (Supplementary Fig. 39) of $Ni/Yb_2O_3$, Ni(111) and $Yb_2O_3$(222) planes are detected as the main crystal faces. This is consistent with the XRD patterns of Ni and $Yb_2O_3$, where Ni(111) and $Yb_2O_3$(222) planes show the strongest diffraction peaks (Supplementary Fig. 7a), respectively. The Ni(111) and $Yb_2O_3$(222) planes are connected in different angles, where the clear phase boundaries are found (Supplementary Fig. 39). In the magnified HRTEM image of $Ni/Yb_2O_3$, the lattice fringes of Ni(111) and $Yb_2O_3$(222) planes are arranged in parallel, between which an interface is observed (Fig. 3e). Line-scanning intensity profile obtained from the blue dashed boxes in Supplementary Fig. 39i enables us to distinguish the Ni and Yb atoms clearly based on their obviously different intensities, that is, the contrast intensity of Ni is much smaller than that of Yb caused by the smaller atomic number of Ni. The distance from a Ni atom to a nearby Yb atom is 0.300 nm (Supplementary Fig. 39i), which is very close to that between two adjacent Yb atoms (0.301 nm). This reveals that there are O atoms between the Ni atom and Yb atom, and thus the existence of Ni–O bonds in the interface between Ni and $Yb_2O_3$.

In addition, the fast Fourier transformation (FFT) pattern (Fig. 3f) also exhibits clear lattice signal of both Ni(111) and $Yb_2O_3$(222) planes, as well as their equivalent planes of Ni($\bar{1}11$) and $Yb_2O_3$($\bar{2}22$). The inversed FFT (IFFT) patterns (Fig. 3g) taken from the selected red and yellow dashed boxes in Fig. 3e reveal a near-parallel relationship of the Ni(111) and $Yb_2O_3$(222) planes, which also illustrates that the $Yb_2O_3$($\bar{2}22$) crystal faces grow along the Ni($\bar{1}11$) faces parallelly. The corresponding schematic structural diagram (Fig. 3h) shows the phase interface of Ni and $Yb_2O_3$ in detail, providing a deep insight into the heterostructure. Moreover, the lattice distance between $Yb_2O_3$($\bar{2}22$) planes (0.301 nm) is nearly one and a half times longer than that between Ni($\bar{1}11$) planes (0.204 nm). This reveals that $Yb_2O_3$($\bar{2}22$) planes can be regularly connected to Ni($\bar{1}11$) planes as highlighted by the red solid lines in Fig. 3h. Beyond the visual TEM images, line-scan electron energy loss spectroscopy (EELS) was taken to clarify the interface structure (Supplementary Fig. 40a), which was recorded along the red arrow in Supplementary Fig. 40b. The obtained spectrum clearly presents the distribution of Ni and its interface with $Yb_2O_3$. The intensity profiles extracted from the EELS spectrum illustrate both Ni L-edge and Yb M-edge signals on interface (II), which demonstrates a tight link between Ni and $Yb_2O_3$ (Supplementary Fig. 40c). In comparison with the bulk-phase Ni (III), a slight positive shift of Ni–O band with a higher peak intensity for the interfacial Ni L-edge peak is observed (Supplementary Fig. 40d), which further confirms the chemical links (i.e. Ni–O bonds) in the interface of $Ni/Yb_2O_3$

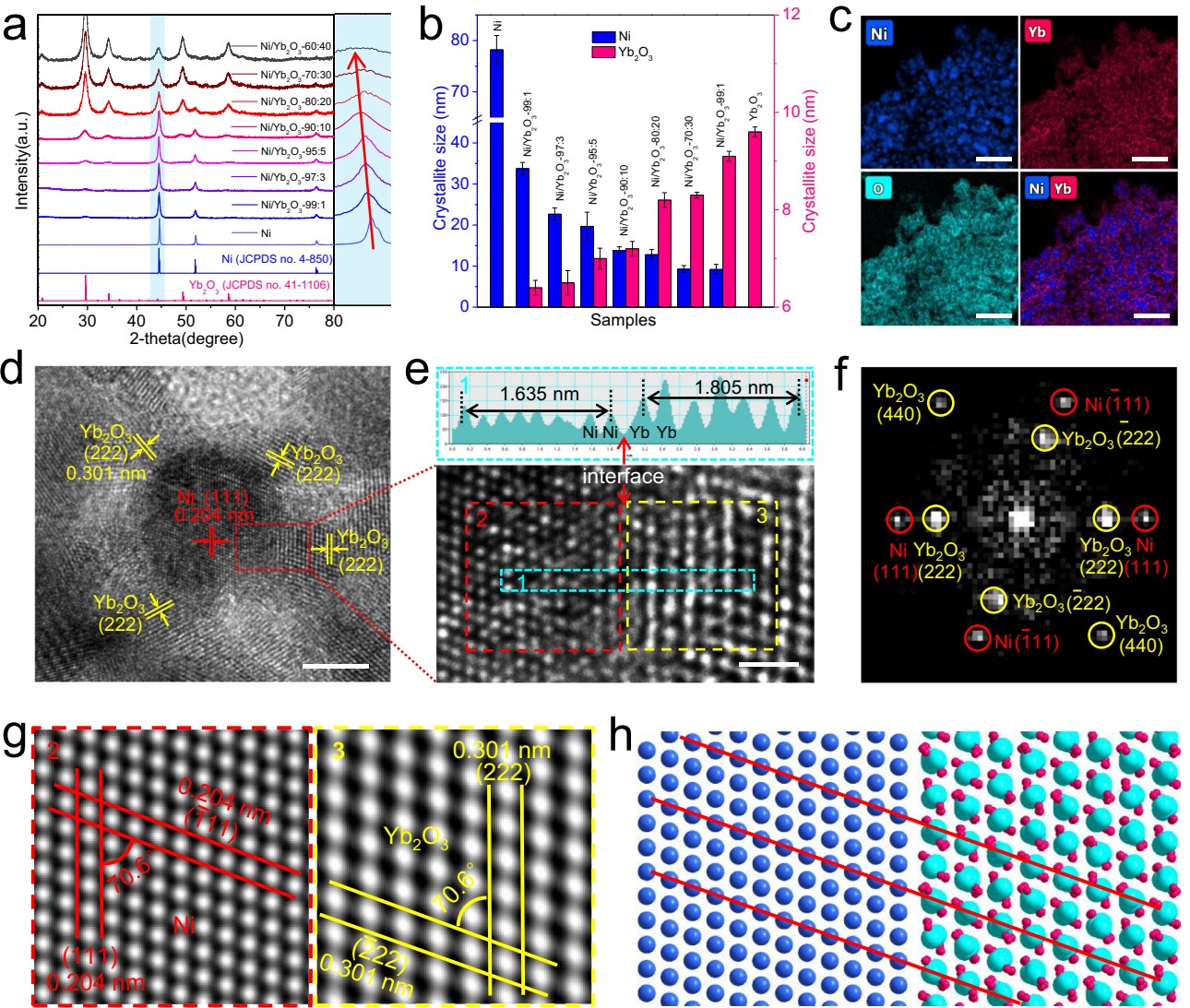

**Fig. 3 Structural characterizations of Ni/Yb₂O₃.** a XRD patterns of Ni and Ni/Yb₂O₃ hybrids with various Ni:Yb molar ratios. b Crystallite sizes derived from XRD patterns with the Debye–Scherrer equation. The error bars show the standard derivation based on triplicate measurements. c Elemental mapping of Ni/Yb₂O₃ (scale bar: 100 nm). d, e HRTEM images of Ni/Yb₂O₃ (scale bar: 5 nm for d and 1 nm for e), and line scan of HRETM image. f Fast Fourier transform (FFT) pattern from (e). g Inverse FFT patterns corresponding to the regions of 2 and 3 in (e). h Schematic diagram of the structures corresponding to the regions of 2 and 3 in (e). The blue, magenta, and cyan spheres represent the Ni, O, and Yb atoms, respectively.

hybrid. For comparison, the Ni/CeO₂ nanoparticle on graphitic plate was also synthesized by the same method (Supplementary Fig. 41).

X-ray absorption spectroscopy (XAS) and XPS were used to explore the impact of coupling Yb₂O₃ on the chemical environments and electronic structures of Ni. Figure 4a presents the X-ray absorption near-edge structure (XANES) spectra of Ni/ Yb₂O₃ at Ni K-edge, which is consistent with that of the pristine Ni and Ni foil reference, revealing the retentive metallic Ni in Ni/ Yb₂O₃. The near-edge adsorption energy of Ni in Ni/Yb₂O₃ shifts to a higher binding energy compared with that of the pristine Ni (Fig. 4a inset), which indicates that the Ni nanoparticles in Ni/ Yb₂O₃ are partly positively charged and the electrons are transferred from Ni to Yb₂O₃. This significant electron transfer also reveals the strong coupling between Ni and Yb₂O₃, which agrees with the strong interfacial contacts between Ni and Yb₂O₃ (Fig. 3e). To further trace the radial structure function around Ni, the extended X-ray absorption fine-structure (EXAFS) spectra of Ni/Yb₂O₃ and Ni were in-depth analyzed. A prominent Fourier

transforms peak of Ni/Yb₂O₃ at 2.41 Å in R space plot is clearly observed for the Ni–Ni path, which is similar to the pristine Ni (Fig. 4b). The results from EXAFS wavelet transform show only one intensity maximum at ca. 8.2 Å⁻¹ in k space, corresponding to the Ni–Ni bond in Ni, which further confirms the metallic state of Ni in Ni/Yb₂O₃ (Fig. 4c). The decrease of Ni–Ni peak intensity in Ni/Yb₂O₃ compared with that in pristine Ni manifests the damped coordination structure of Ni (Fig. 4b)[49]. The Ni K-edge EXAFS fitting (Supplementary Table 4) indicates that the first-shell Ni–Ni coordination numbers (CNs) reduce from Ni to Ni/ Yb₂O₃. The lower CN can be ascribed to the smaller crystal sizes and rich surface steps of Ni nanoparticles in the hybrid[50], which can increase the catalytic active sites and adjust the adsorption ability of Ni/Yb₂O₃. The normalized Yb L-edge XANES spectrum for Ni/Yb₂O₃ is consistent with that for the as-synthesized Yb₂O₃ (Fig. 4d). Also, the results from FT-EXAFS and EXAFS wavelet transform suggest that the Yb component in Ni/Yb₂O₃ possesses a similar coordination environment to that in pristine Yb₂O₃ (Fig. 4e, f).

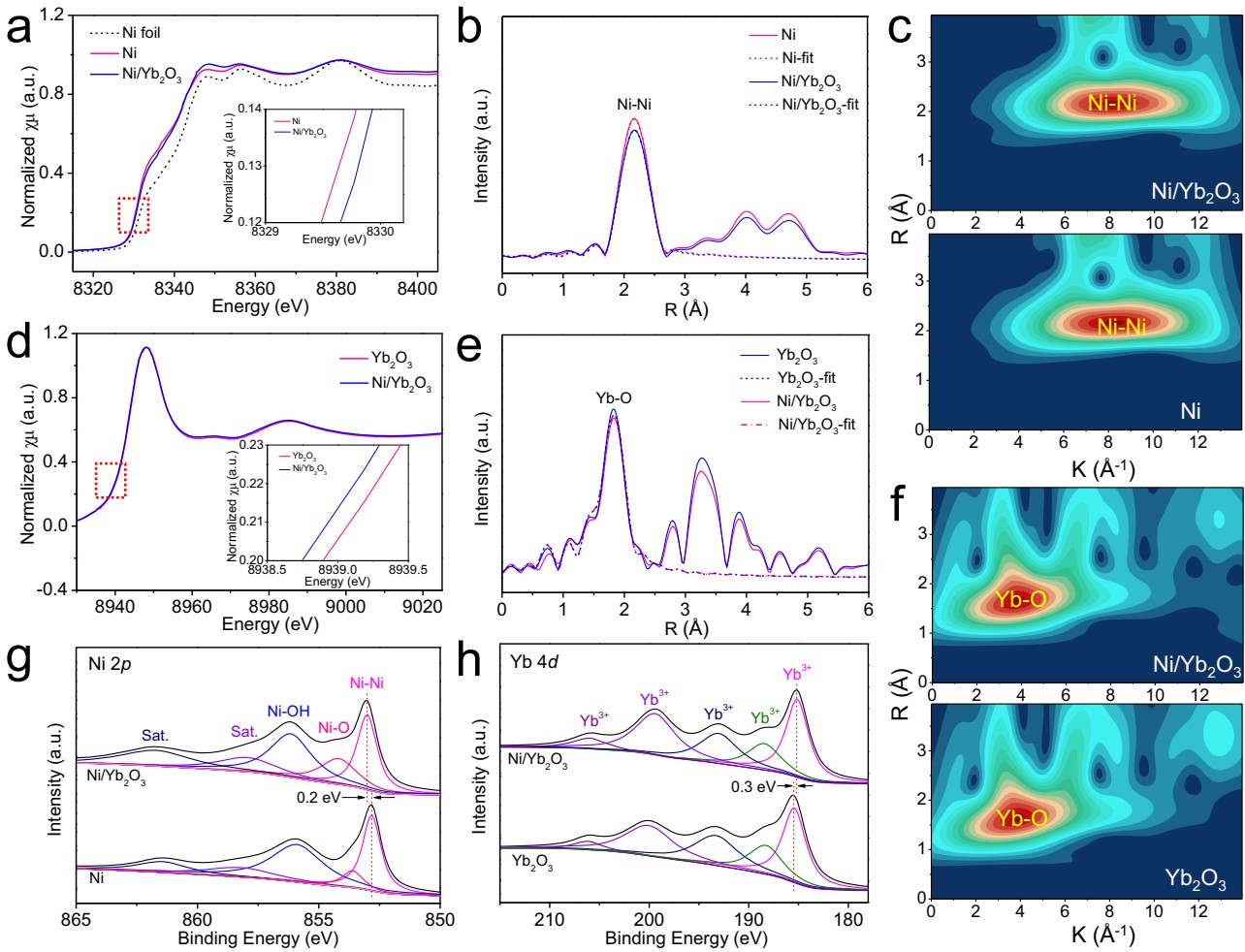

**Fig. 4 Spectroscopic characterizations of Ni/Yb₂O₃ hybrid. a** Ni K-edge XANES spectra of Ni/Yb₂O₃, pristine Ni, and Ni foil reference. **b**, **c** Fourier transforms and wavelet transforms of EXAFS spectra for Ni/Yb₂O₃ and pristine Ni. **d** Yb L-edge XANES spectra of Ni/Yb₂O₃ and pristine Yb₂O₃. **e**, **f** Fourier transforms and wavelet transforms of EXAFS spectra for Ni/Yb₂O₃ and pristine Yb₂O₃. **g** Ni 2p XPS spectra of Ni/Yb₂O₃ and pristine Ni. **h** Yb 4d XPS spectra of Ni/Yb₂O₃ and pristine Yb₂O₃.

As observed in Ni 2p XPS spectra (Fig. 4g), in addition to the Ni–Ni bands, there are Ni–O bonds in Ni/Yb₂O₃ and pristine Ni, which is different from that no distinct first-shell of Ni–O is observed in the Ni K-edge FT-EXAFS and EXAFS wavelet transform. This is primarily because that XPS is a surface sensitive analytical technique whereas XAS in the configuration used in these experiments is a bulk technique[7]. The peak intensity of Ni–O bond (Fig. 4g) in Ni/Yb₂O₃ is obviously higher than that in metallic Ni, which illustrates the presence of Ni–O interactions between Ni and Yb₂O₃[51,52]. The positively shifted Ni 2p peaks of Ni (Fig. 4g) and the negatively shifted Yb 4d peaks (Fig. 4h) in Ni/Yb₂O₃ further illustrate the strong electronic interactions between Ni and Yb₂O₃ in interface. All these results indicate that the introduction of Yb₂O₃ can modulate the geometric and electronic structures of Ni in Ni/Yb₂O₃ hybrid, which plays a significant role on its enhanced electrocatalytic activity of HER.

**Evaluation of electrocatalytic activity and stability for Ni/Yb₂O₃.** Among the Ni/Yb₂O₃ electrodes with different compositions, Ni/Yb₂O₃—90:10 shows the best performances in terms of overpotential, Tafel slope, and ECSA-based specific activity (Supplementary Figs. 42–44). It can be attributed to its large ECSA, high intrinsic electrocatalytic activity, and high conductivity. As discussed above, the high ECSA of Ni/Yb₂O₃—90:10

results from its appropriate amount of Yb₂O₃. And for the intrinsic catalytic activity, incorporating oxophilic Yb₂O₃ into metallic Ni affords efficient dual active sites for both H₂O dissociation and H₂ formation. Nevertheless, the challenge to achieve the best activity of the Ni/Yb₂O₃ hybrids is that an optimal balance of H₂O dissociation rate and H₂ formation rate is needed to accelerate the overall HER kinetics through steering the proportion of Ni and Yb₂O₃ components (Supplementary Fig. 45). As illustrated in Supplementary Fig. 45a, H₂O is first adsorbed on the oxophilic Yb₂O₃ in the interface, and then easily broken up into the OH and H intermediates. Then, the adsorbed H intermediate will form H₂ on the Ni sites through the Heyrovsky or Tafel step. As a result, the Ni/Yb₂O₃ heterosurfaces synergistically boost the Volmer step and the subsequent Heyrovsky or Tafel step of alkaline HER. However, overmuch Yb₂O₃ component will lead to insufficient Ni sites for H₂ formation, which also will result in excessive OH intermediate to limit the H₂O adsorption. On the contrary, if the Yb₂O₃ component is too less, water dissociation (i.e. Volmer step) becomes a rate-limiting step, leading to the insufficient rate of H_{ads} formation. As shown in Supplementary Fig. 42b, the HER kinetics for the Ni/Yb₂O₃ hybrids is consistent with their catalytic activities, confirming that the alkaline HER activity on Ni/Yb₂O₃ is highly dependent on its proportion of Ni:Yb₂O₃. In addition, the introduction of Yb₂O₃ in metal Ni can lead to a marked reduction

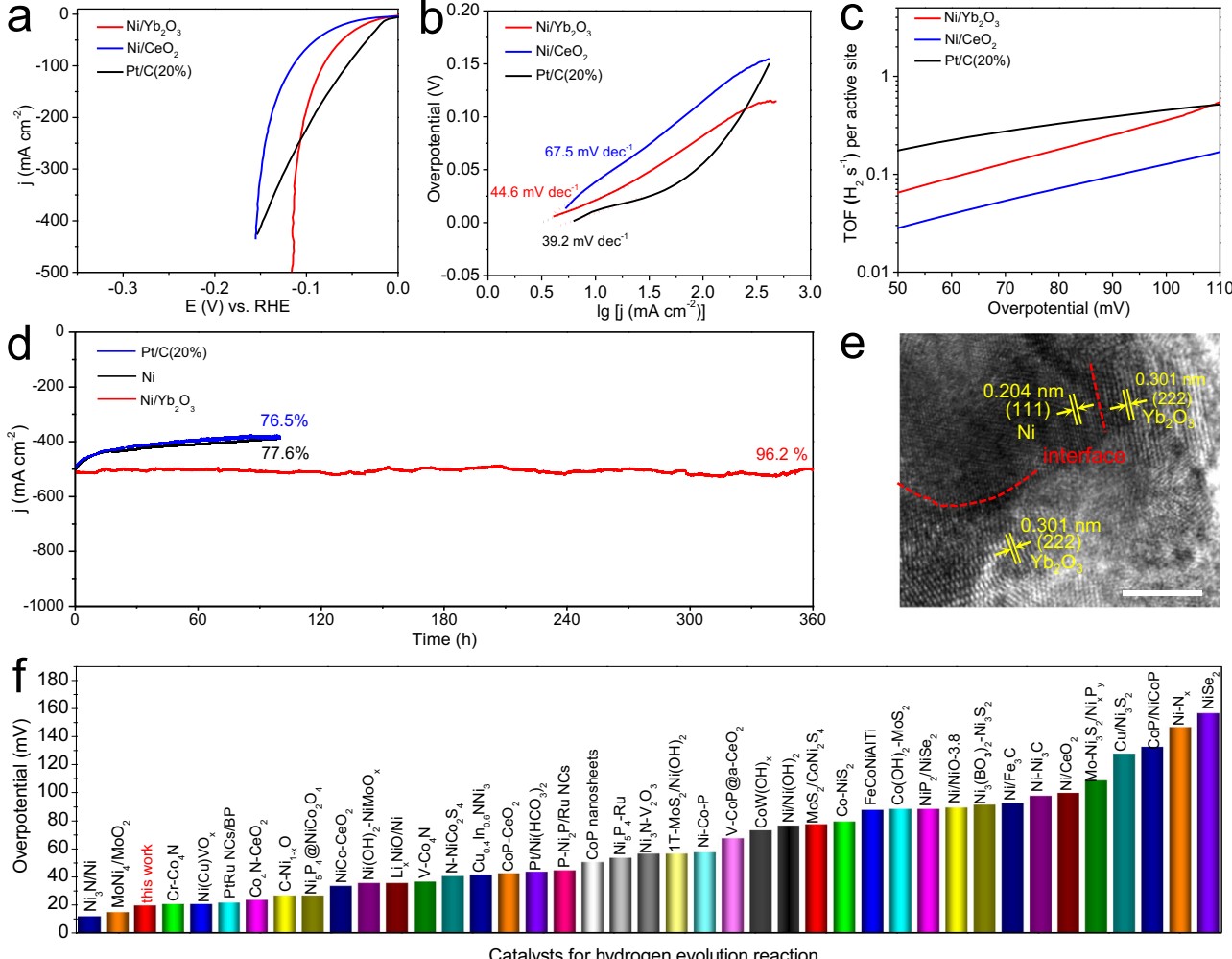

**Fig. 5 Electrocatalytic HER activity and stability for Ni/Yb$_2$O$_3$ electrode in 1.0 M KOH. a** Polarization curves (scan rate: 5 mV s$^{-1}$) of Ni/Yb$_2$O$_3$, Ni/CeO$_2$, and Pt/C(20%) electrodes with a mass loading of ca. 3.5 mg cm$^{-2}$. **b** Tafel plots derived from the curves in (**a**). **c** TOF values of Ni/Yb$_2$O$_3$, Ni/CeO$_2$, and Pt/C(20%) electrodes. **d** Chronopotentiometric curves of Ni/Yb$_2$O$_3$, Ni, and Pt/C(20%) electrodes at the overpotential of 116.0, 305.0, and 167.0 mV, respectively. **e** TEM image of Ni/Yb$_2$O$_3$ after the HER test. **f** Comparison of the HER activities for Ni/Yb$_2$O$_3$ and the reported electrocatalysts (Supplementary Table 6).

in conductivity of the Ni/Yb$_2$O$_3$ hybrids, caused by the very poor conductivity of Yb$_2$O$_3$ (Supplementary Fig. 46). Excessive Yb$_2$O$_3$ component will impede the electron transfer during the HER. Therefore, the 90:10 is the best molar ratio of Ni:Yb in the Ni/Yb$_2$O$_3$ hybrids.

The high alkaline HER activity of Ni/Yb$_2$O$_3$ was further evaluated by comparing with those of Ni/CeO$_2$ and Pt/C(20%). To reach the current density of 10 mA cm$^{-2}$, Ni/Yb$_2$O$_3$ has a small overpotential requirement of 20.0 mV (Fig. 5a and Supplementary Fig. 47). This overpotential is much lower than that of Ni/CeO$_2$ electrode (41.1 mV) and only 10.0 mV higher than that of the benchmark Pt/C(20%) electrode (Supplementary Table 5). Notably, the overpotential of Ni/Yb$_2$O$_3$ at large current density is lower than that of Pt/C(20%), suggesting its higher HER activity. This is mainly because Ni/Yb$_2$O$_3$ electrode not only has high intrinsic activity but also possesses hydrophilic self-supported electrode structure, which could ensure fast electron and mass transport at large current density[53]. As shown in Fig. 5b, the Tafel slope of Ni/Yb$_2$O$_3$ (44.6 mV dec$^{-1}$) is lower than that of Ni/CeO$_2$ (67.5 mV dec$^{-1}$) and is close to that of Pt/C(20%) (39.2 mV dec$^{-1}$). For Pt/C(20%) electrode, the poor contact between the physically coated Pt/C(20%) catalyst and

substrate (Supplementary Fig. 48) results in mass transfer limit at large current density, which is certified by the upward deviation at high overpotential in its Tafel plot[54]. With high catalytic activity, Ni/Yb$_2$O$_3$ only needs 116.0 mV to attain a high current density of 500 mA cm$^{-2}$. Moreover, the TOFs of Ni/Yb$_2$O$_3$ measured from 50 to 110 mV overpotentials are higher than those of Ni/CeO$_2$ (Fig. 5c). Notably, the TOF value of Ni/Yb$_2$O$_3$ (0.362 H$_2$ s$^{-1}$) is over 3 times higher than that of Ni/CeO$_2$ (0.120 H$_2$ s$^{-1}$) at 100 mV, which confirms that Yb$_2$O$_3$ is a better promoter of Ni catalyst for alkaline HER relative to CeO$_2$. In addition, the Faradaic efficiency of HER for Ni/Yb$_2$O$_3$ catalyst is nearly 98% (Supplementary Fig. 49).

Besides activity, stability of electrocatalysts at high current density is a critical criterion for the practical application. To evaluate the HER durability of Ni/Yb$_2$O$_3$, the continuous CV sweep was measured from 0 to −0.35 V, with Ni as a contrast. After 5000 cycles, the polarization curve of Ni shows a significant change, while Ni/Yb$_2$O$_3$ keeps the initial activity (Supplementary Fig. 50). The long-term chronoamperometry curves were also taken at an overpotential of 116.0 mV for Ni/Yb$_2$O$_3$, 305.0 mV for Ni, and 167.0 mV for Pt/C(20%). The Ni/Yb$_2$O$_3$ electrode shows excellent stability at high current density of ~500 mA cm$^{-2}$ for

360 h, while Ni and Pt/C(20%) exhibit a rapid current decay after 100 h water electrolysis (Fig. 5d). In addition, the HER performance of Ni/Yb$_2$O$_3$ was also tested by using Hg/HgO as the reference electrode (Supplementary Fig. 51), which further confirms its high catalytic activity and stability for HER. To verify whether oxygen vacancies in Yb$_2$O$_3$ affect the electrocatalytic activity and stability, the Ni/Yb$_2$O$_3$ hybrids (Ni/Yb$_2$O$_3$—1.0 h, Ni/Yb$_2$O$_3$—2.0 h, and Ni/Yb$_2$O$_3$—4.0 h) with different oxygen vacancy concentrations were tested (Supplementary Fig. 52). The results illustrate that the three hybrids show almost identical catalytic activity and stability for alkaline HER, revealing that the performance of Ni/Yb$_2$O$_3$ hybrid are independent to the oxygen vacancies in Yb$_2$O$_3$. In terms of the overpotential at 10 mA cm$^{-2}$ and Tafel slope, the Ni/Yb$_2$O$_3$ hybrid not only outperforms most of the Ni-based HER electrocatalysts, but also precedes most of reported alkaline HER electrocatalysts (Fig. 5f and Supplementary Table 6), manifesting its respectable catalytic activity.

The high durability of Ni/Yb$_2$O$_3$ was also confirmed by post-electrolysis characterization (Supplementary Fig. 53). The XRD pattern for Ni/Yb$_2$O$_3$ after a long-term stability test matches with the initial status before test (Supplementary Fig. 53a). The Ni/Yb$_2$O$_3$ nanoparticles are still attached onto the base tightly without morphology change (Supplementary Fig. 53b). The TEM image reveals that Ni/Yb$_2$O$_3$ remains the heterojunction structure with Yb$_2$O$_3$ decorating on the surface of Ni nanoparticles (Fig. 5e). Its corresponding element mapping illustrates the uniform distribution of Ni, Yb, and O after 360 h test (Supplementary Fig. 53c, 53d). While for Ni, an obvious oxide layer is generated on the surface of Ni nanoparticles, resulting in a Ni/NiO$_x$ core–shell structure as indicated by the TEM image, in spite of the unchanged morphology and bulk phase structure of the Ni particles (Supplementary Figs. 54a–c). The almost invisible metallic Ni content and the dominant NiO$_x$ in the XPS spectrum of Ni element also illustrate the severe oxidation of Ni (Supplementary Fig. 54d), resulting in the loss of active sites for HER, thereby the degradation of catalytic activity. As reported in previous literatures, this is an inevitable and ubiquitous problem for metallic Ni HER electrocatalysts under alkaline conditions[5,7,9]. Notably, Ni/Yb$_2$O$_3$ shows the less oxidation of Ni nanoparticles after the 360 h electrolysis operation, as illustrated by the Ni 2$p$ XPS spectrum (Supplementary Fig. 53e). Moreover, the unaltered ECSA after HER test also indicates the high stability of Ni/Yb$_2$O$_3$ electrode (Supplementary Fig. 55). It is widely accepted that water and dissolved oxygen in the electrolyte play a major role on the corrosion of metallic Ni. Since Yb$_2$O$_3$ is highly stable under the pH and potential ranges of the HER tests, the surface anchored Yb$_2$O$_3$ can serve as the protection shell for Ni phase, preventing the oxidation of Ni phase[10,55]. Moreover, electronic interaction between Ni and Yb$_2$O$_3$ can also decrease the adsorption energy of O$_2$ on the Ni sites, thereby further relieving the oxygen corrosion. These facts are beneficial to the electrochemical stability of the Ni/Yb$_2$O$_3$ hybrid for HER at high current density. As a result, the synchronous enhancement of HER activity and stability of Ni-based materials can be achieved by coupling the Yb$_2$O$_3$ promoter. To further illustrate the interfacial effect between Ni and Yb$_2$O$_3$, a Ni + Yb$_2$O$_3$ sample was prepared by mechanically mixing Ni and Yb$_2$O$_3$ powder using Nafion as the binder. Obviously, the Ni + Yb$_2$O$_3$ catalyst shows quite inferior activity and stability compared with that of Ni/Yb$_2$O$_3$ (Supplementary Fig. 56), which reveals that the strong coupling interface between Ni and Yb$_2$O$_3$ plays a key role in the enhanced catalytic activity and stability of Ni/Yb$_2$O$_3$[56].

**Theoretical simulations**. The first principle calculations were used to elucidate the theoretical enhancement of intrinsic HER

activity and stability for the heterogeneous interface in Ni/Yb$_2$O$_3$ as compared with pristine Ni. First, the structural models of Ni/Yb$_2$O$_3$, pristine Ni, Yb$_2$O$_3$, and Ni/CeO$_2$ were established based on the determined structures of these materials (Supplementary Fig. 57). The energy barrier of water dissociation is a critical factor to characterize the intrinsic catalytic activity for HER in alkaline media[3,57]. As proved by the CV curves for OH adsorption and desorption experiments (Supplementary Fig. 25), the incorporation of oxophilic lanthanide oxides in metallic Ni strengthens the adsorption energy of OH. This is also verified by the first principle calculation result, that is, the adsorption energy of OH on Ni(111)/Yb$_2$O$_3$(222) is more negative than that on pure Ni (Supplementary Fig. 58). The strong adsorption of OH on Ni(111)/Yb$_2$O$_3$(222) indicates the favorable adsorption of water molecules and cleaving of HO–H bond. To validate this prediction, the energy barriers for water dissociation on catalysts were taken by density functional theory (DFT) calculation. As shown in Supplementary Fig. 59, the stronger H$_2$O adsorption on Ni/Yb$_2$O$_3$ and Ni/CeO$_2$ hybrids relative to pristine Ni further certifies that the coupling of oxophilic Yb$_2$O$_3$ and CeO$_2$ on Ni significantly promotes the adsorption of water molecules, which will expedite the water dissociation thereon[58]. The H$_2$O dissociation reaction on pure Ni surface, Yb$_2$O$_3$ surface, and interface of Ni/Yb$_2$O$_3$ and Ni/CeO$_2$ were also calculated (Fig. 6a–c and Supplementary Figs. 60–63). With regard to Ni(111)/Yb$_2$O$_3$(222) interface (Fig. 6b and Supplementary Fig. 62), the oxygen of water is absorbed on Yb of Yb$_2$O$_3$ and then the water molecule is broken up to the hydroxyl and hydrogen intermediates, which are adsorbed by Yb and nearby Ni atoms, respectively. As expected, the energy barrier for water dissociation on the interface of Ni(111)/Yb$_2$O$_3$(222) is 0.47 eV (Fig. 6c), which is dramatically lower than those on Ni(111) surface (0.62 eV) and Yb$_2$O$_3$(222) surface (1.12 eV). This result demonstrates that the sluggish water dissociation step on Ni can be greatly facilitated by coupling with Yb$_2$O$_3$, which is consistent with the Tafel slope and EIS analysis. More importantly, this energy barrier is even lower than that of Pt surface (0.56 eV)[59], and is also close to that (0.41 eV) of Ni(111)/CeO$_2$(111) interface. Thus, in addition to the well-known water dissociation promoter CeO$_2$, the bixbyite-type Yb$_2$O$_3$ with suitable oxophilicity is also a promising promoter for water dissociation. The accelerated water dissociation step of Volmer process on Ni(111)/Yb$_2$O$_3$(222) provides enough hydrogen intermediate to the active Ni sites for subsequent Heyrovsky step or Tafel step.

Except for the energy barrier of water dissociation, the free adsorption energy of H* ($\Delta G_{H*}$) is another important descriptor to characterize the alkaline HER activities of electrocatalysts. High-efficiency HER electrocatalysts should possess moderate H* adsorption energy[60]. As for the charge density difference (Fig. 6d), the increased charge densities are clearly represented at the Ni/Yb$_2$O$_3$ interface. This implies strong synergistic interactions between Ni and Yb$_2$O$_3$ in hybrid, which play a vital role in promoting the electron transfer. The differential charge density analysis also reveals that the electron transfer occurs from Ni to O in the Ni/Yb$_2$O$_3$ interface, which thus renders the lowered $d$-band center of the interfacial Ni atom in Ni/Yb$_2$O$_3$ (Fig. 6e) and reduces the strong adsorption energy of H on metallic Ni[61]. Figure 6f shows the calculated $\triangle G_{H*}$ on bare Ni(111), bare Yb$_2$O$_3$(222), Ni(111)/Yb$_2$O$_3$(222) and Ni(111)/CeO$_2$(111) with most energetically stable configurations (Supplementary Fig. 64). With regard to pristine Ni and Yb$_2$O$_3$, the $\triangle G_{H*}$ are calculated to be −0.38 and −2.43 eV, respectively, which indicate the strong adsorption of H on these sites. This will prevent the H* desorption and H$_2$ generation, resulting in the poor HER reaction kinetics[61]. As anticipated, coupling Ni with Yb$_2$O$_3$ significantly optimizes the $\triangle G_{H*}$ of Ni (−0.26 eV). The reduced but

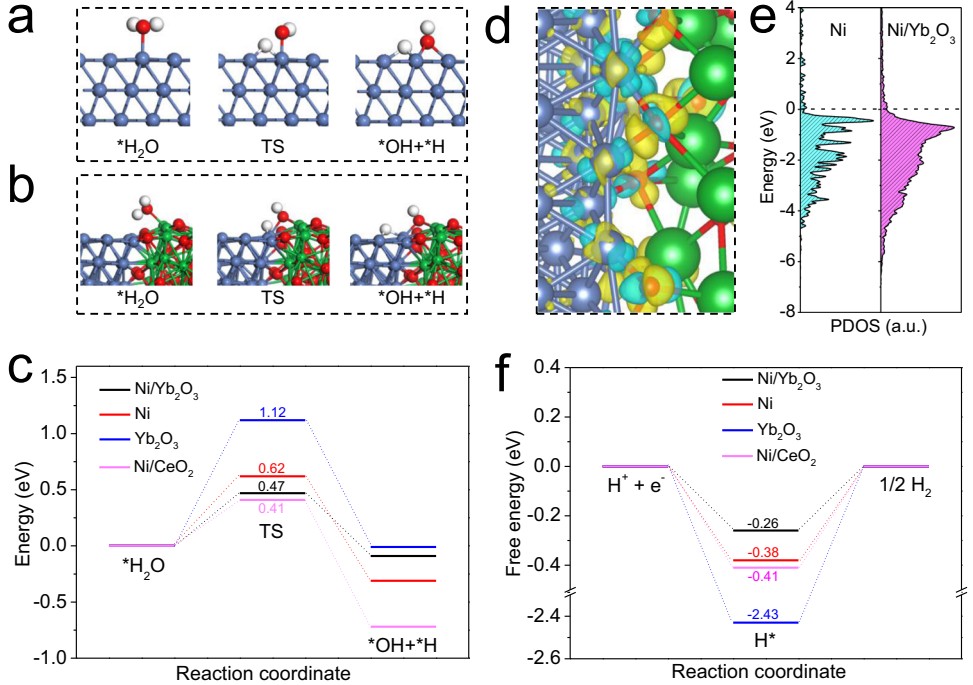

**Fig. 6 Theoretical simulations. a** Atomic configurations of simulated H$_2$O dissociation process on the optimized sites of pristine Ni(111) surface. **b** Atomic configurations of simulated water dissociation process on the optimized sites of Ni(111)/Yb$_2$O$_3$(222) interface. **c** Kinetic barrier of water dissociation on Ni(111)/Yb$_2$O$_3$(222), Ni(111), Yb$_2$O$_3$(222) and Ni(111)/CeO$_2$(111). **d** Difference of charge density for Ni(111)/Yb$_2$O$_3$(222) with the isosurface = 0.004 e bohr$^{-3}$ (yellow and cyan shadows show electron accumulation and electron depletion, respectively). **e** Partial density of states (PDOS) of Ni in pristine Ni and Ni/Yb$_2$O$_3$. **f** Calculated $\Delta G_{H^*}$ for Ni(111)/Yb$_2$O$_3$(222), Ni(111), Yb$_2$O$_3$(222), and Ni(111)/CeO$_2$(111) systems. The blue, red, white, and green spheres represent the Ni, O, H, and Yb atoms, respectively.

optimized H binding energy of Ni/Yb$_2$O$_3$ would favor the transformation of H* to H$_2$, and also expedite the H$_2$ desorption to refresh the catalytic active sites. It is worth noting that the doped CeO$_2$ is inferior in optimizing the H binding energy of Ni and the Ni/CeO$_2$ hybrid still shows the strong H binding energy that is similar to the pristine Ni. This impedes the subsequent Heyrovsky or Tafel step, although the lowered energy barrier of water dissociation (Volmer step) is obtained on the Ni(111)/CeO$_2$(111) hybrid. Remarkably, coupling Yb$_2$O$_3$ with Ni can concurrently lower the H$_2$O-dissociation energy barrier and optimize the $\triangle G_{H^*}$, thereby promoting the kinetics of HER in alkaline medium as experimentally observed.

In fact, there are two kinds of Ni sites in the Ni/Yb$_2$O$_3$ hybrids, including the interfacial Ni sites coupling with Yb$_2$O$_3$ and the other Ni sites far from interfaces (Supplementary Fig. 45a). Due to the lack of effective sites for water dissociation, the energy barriers of water dissociation on the Ni sites far from Ni/Yb$_2$O$_3$ interface (0.61 eV) is almost identical to that of pure Ni (0.62 eV), which is clearly higher than that (0.47 eV) of the Ni/Yb$_2$O$_3$ interface (Supplementary Figs. 65 and 66a). The $\triangle G_{H^*}$ on the Ni sites far from Ni/Yb$_2$O$_3$ interface (−0.39 eV) is also similar to that (−0.38 eV) on pure Ni (Supplementary Figs. 66b and 67), since the far distance limits the electronic interactions between these non-interface Ni sites and Yb$_2$O$_3$[62]. This result indicates that the Ni site far from Ni/Yb$_2$O$_3$ interface has low catalytic activity and the heterogeneous interface is the catalytic active center of Ni/Yb$_2$O$_3$, on which the alkaline HER occurs preferentially and rapidly. Additionally, the Ni sites far from Ni/CeO$_2$ interface also display the analogous energy barriers of water dissociation and $\triangle G_{H^*}$ with pure Ni (Supplementary Figs. 66–68), further confirming their low activity.

Generally, metallic Ni is easily subject to oxidation by oxygen dissolved in the electrolyte or oxygen migrated from the counter

electrodes, resulting in the loss of active sites. The stability experiments and the post-electrolysis characterizations verify that Yb$_2$O$_3$ coupling could relieve the oxidation corrosion of Ni, thereby improving the stability of Ni/Yb$_2$O$_3$ for catalyzing HER. Furthermore, the DFT calculations illustrate that the adsorption energies of O$_2$ on different Ni adsorption sites of Ni/Yb$_2$O$_3$ are much weaker than that on bare Ni surface (Supplementary Fig. 69). This suggests that the Ni phase in Ni/Yb$_2$O$_3$ is more resistant to O$_2$ interaction and oxidation erosion than pure Ni[63], which ensures the highly active heterojunction of Ni and Yb$_2$O$_3$ during the HER process. Moreover, the lowered H binding energy on Ni of Ni/Yb$_2$O$_3$ can decrease the hydrogen-adsorption poison of Ni active sites, thus improving its long-term stability for HER[64].

## Discussion
In summary, the enhancement effect of bixbyite-type lanthanide sesquioxides for alkaline HER performances has been proposed and validated based on the designed Ni/Ln$_2$O$_3$ model catalysts. The screened Ni/Yb$_2$O$_3$ not only exhibits the best HER catalytic performances in the diverse Ni/Ln$_2$O$_3$ hybrids, but also outperforms the well-known Ni/CeO$_2$ electrocatalyst, revealing that Yb$_2$O$_3$ is a better performance enhancer of Ni for alkaline HER relative to the traditional CeO$_2$. Incorporating oxophilic Yb$_2$O$_3$ into metallic Ni affords the dual active sites, greatly accelerating the dissociation of water, and the localized electronic polarization between Ni and Yb$_2$O$_3$ optimizes the hydrogen adsorption energy, thus boosting the overall HER kinetics. In addition, the tightly coupled Yb$_2$O$_3$ with high chemical stability significantly lowers the grain sizes and inhibits the chemical oxidation corrosion of Ni, resulting in enlarged ECSA and robust stability. Remarkably, the Ni/Yb$_2$O$_3$ electrode exhibits an ultralow overpotential of 20.0 mV at 10 mA cm$^{-2}$ and retains the high stability

over 360 h at a large current density of 500 mA cm$^{-2}$, preceding most of the reported alkaline HER catalysts. The high activity and durability endow the Ni/Yb$_2$O$_3$ electrode with great potentials in large-scale application for industrial electrolyzer. More significantly, the ability of Ln$_2$O$_3$ to promote the water dissociation should not be limited to alkaline HER system, which will be also available to other catalytic reactions involving water dissociation. The applications of these promising Ln$_2$O$_3$ promoters toward those electrocatalytic processes, such as CO$_2$RR, NRR, ORR, and water–gas shift (WGS), are underway.

## Methods

**Preparation of materials**. For the preparation of a Ni/Ln$_2$O$_3$ electrode, the graphite plate ($1 \times 2$ cm$^2$) was cleaned in ethanol, dilute HCl and ultrapure water, respectively, then dried at room temperature. The precursor was prepared by the electrodeposition method in an electrolytic cell with graphite plate as working electrode. The deposited electrolyte is a water solution of 0.09 M Ni(NO$_3$)$_2$ and 0.01 M Ln(NO$_3$)$_3$. During the electrodeposition process of Ni(OH)$_2$/Ln(OH)$_3$, the GP was treated at 20 mA cm$^{-2}$ for 600 s and then $-40$ mA cm$^{-2}$ for 600 s. Subsequently, the deposited Ni(OH)$_2$/Ln(OH)$_3$ was converted to Ni/Ln$_2$O$_3$ in a tube furnace at 500 °C under 10% H$_2$/Ar mixture for 4 h with a heating rate of 5 °C min$^{-1}$. For comparison, Ni/Yb$_2$O$_3$ with different Ni:Yb molar ratios (i.e. 99:1, 97:3, 95:5, 90:10, 80:20, 70: 30, and 60:40) were synthesized by adjusting the amount of Ni(NO$_3$)$_2$ and Yb(NO$_3$)$_3$. In addition, the Ni and Ln$_2$O$_3$ electrodes were prepared by a similar method, using only 0.1 M Ni(NO$_3$)$_2$ or 0.1 M Ln(NO$_3$)$_3$, respectively. The Ni/CeO$_2$ electrode was also prepared by this method for a comparison. The catalyst loading for each electrode is ca. 3.5 mg cm$^{-2}$.

**Characterizations of materials**. Powder XRD measurements were taken on a Rigaku model Ultima IV diffractometer with Cu-Kα X-ray source. SEM images were collected on a FEI Nova Nano 230 scanning electron microscope. TEM equipped with EDS and SAED was conducted on a Tecnai G$^2$ F20 electron microscope. XPS was measured by a Kratos Axis Ultra DLD spectrometer. Inductively coupled plasma atomic emission spectrometer (ICP-AES) was performed on PerkinElmer Optima 83000. X-ray absorption fine structure spectra (Ni K-edge/Yb L-edge) were collected at BL14W beamline in Shanghai Synchrotron Radiation Facility (SSRF). The storage rings of SSRF were operated with a stable current of 200 mA at 3.5 GeV. With Si(111) double-crystal monochromator, the data collection was taken in Transmission mode using Lytle detector under ambient conditions.

**Electrochemical measurements**. Electrochemical measurements were performed in a three-electrode cell with a volume of 150 mL using the Bipotentiostat workstation (Pine Research Instrumentation, Basic Wave Driver 20 Bundle, USA) and Solartron ModuLab XM. The as-prepared self-supported electrode was used as the working electrode. The area of as-prepared electrode is 2.0 cm$^2$, of which the part loading catalyst immersed into the electrolyte is 1.0 cm$^2$ (Supplementary Fig. 2). The Pt/C(20%)@GP electrode was prepared by drop casting Pt/C(20%) catalyst ink on GP (3.5 mg cm$^{-2}$). The ink was achieved by ultrasonically dispersing 10 mg Pt/C(20%) in a mixed solution containing 950 μL ethanol and 50 μL Nafion solution. A 1.0 M KOH solution purged with Ar gas was applied as the electrolyte. The LSV from 0.1 to $-0.5$ V were recorded at a rate of 5 mV s$^{-1}$. EIS measurements were taken in the frequency region from 100 kHz to 0.01 Hz. The amounts of produced H$_2$ were collected using the drainage route. The Faradic efficiency was defined as the ratio of experimental H$_2$ production amount to theoretical H$_2$ production amount. The RHE calibration of reference electrode was taken in H$_2$-saturated 1 M KOH using a platinum plate as the working electrode (Supplementary Figs. 51a, 70). All potentials were reported relative to the RHE scale unless noted. All polarization curves were iR-corrected using: $E_{iR} = E_{tested} - i \times 0.8 R_s$ ($R_s$ is resistance of system).

**Computational methods**. The DFT calculations were performed by Vienna Ab-initio Simulation Package (VASP)[65]. The ion–electron interaction was described by using the projector-augmented wave (PAW) method[66]. The electron exchange-correlation was revealed by the functions of Perdew, Burke, and Ernzerhof (PBE) of generalized gradient approximation (GGA)[67]. The DFT-D3 method was taken to describe the van der Waals correction. For all calculations, the cutoff energy was set to be 520 eV. Furthermore, the periodic boundary conditions with the vacuum slab of 15 Å were used to avoid the interactions between neighboring periodic structures. A $3 \times 3 \times 1$ Monkhorst–Pack grid was used for all calculations[68]. The convergence threshold for the geometry optimization was set to $10^{-5}$ eV in energy and 0.01 eV Å$^{-1}$ in force, respectively. The convergence threshold in force for transition state searching calculations was set to 0.05 eV Å$^{-1}$. The climbing image nudged elastic band (CI-NEB) method was taken to search the minimum energy pathway (MEP) for water dissociating into OH and H radical[69]. The H$_2$O

absorption energy was calculated by Eq. (3):

$$\Delta E_{H_2O} = E_{(surf+H_2O)} - E_{(surf)} - E_{H_2O} \qquad (3)$$

where $E_{(surf+H_2O)}$ and $E_{(surf)}$ are the energies of the surfaces with and without H$_2$O adsorbate, respectively. $E_{H_2O}$ is the energy of a H$_2$O molecule. The free energy change ($\Delta G$) of adsorbate is calculated according to Eq. (4):

$$\Delta G = \Delta E + \Delta E_{ZPE} - T\Delta S \qquad (4)$$

in which $\Delta E$ is DFT calculated total energy change, $\Delta E_{ZPE}$ is the zero-point energy change, and $\Delta S$ is the entropy difference.

## Data availability

All data generated in this study are provided in the Supplementary Information/Source Data file. Source data are provided with this paper.

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

## Acknowledgements
This work was supported by the National Natural Science Foundation of China (52101268, H.-S.) and the Science & Technology Development Fund of Tianjin Education Commission for Higher Education (2019KJ088, H.-S.).

## Author contributions
H.-S. and M.-D. proposed the concept. H.-S., C.P.-L. and M.-D. directed the research. C.-T. and C.-L. completed most of the experiments. Z.-Y. contributed to the X-ray absorption fine structure spectroscopy and DFT calculations. X.-F., R.-H., Y.-L. and J.-C. conducted the synthesis and characterization of partial catalysts. Z.-Z. assisted with the structural characterizations of catalysts. H.-S., Z.-Y. and M.-D. co-wrote the manuscript. All authors participated in data analysis and manuscript discussion.

## Competing interests
The authors declare no competing interests.
