## [Peer Review File · Nature Communications]

Bixbyite-type Ln₂O₃ as promoters of metallic Ni for alkaline electrocatalytic hydrogen evolutionREVIEWER COMMENTS

Reviewer #1 (Remarks to the Author):

The paper by Sun et al. deals with the Bixbyite-type Ln_2O_3 as remarkable promoters of metallic Ni for alkaline electrocatalytic hydrogen evolution. This work is a very good contribution to the field and could be published in Nature Communications after minor revision as mentioned below:

1. CAS number, purities and providers of all chemicals (including the used gases) should be added in the experimental section.
2. Relative error should be added to all values, tables and figures used in the manuscript (and not some of them). This error should determine the number of digits used after the decimal point. Is it relevant for instance to use 2 numbers after the decimal point for measurement realized by EDX?
3. EDX should be redrawn using origin. The table should be removed from inset

Reviewer #2 (Remarks to the Author):

This manuscript reports a new type of catalyst where Ni nanoparticles are mixed with a small proportion of lanthanide oxides and tested for the hydrogen evolution reaction. Interestingly, it is proved that some members of the series Ln_2O_3 are very well suited for catalysts of HER. The paper is based on a series of experiments and first principles computations. The outcome is interesting and potentially very useful. I would be in favor of publication, but would like to see some clarifications of the following points.

1. It is difficult to understand what are the active sites in the mixed catalyst. From the computations it is suggested that the interfaces between Ni and the oxide is where the reaction may take place. However, why is the number of active sites enhanced for a >90% Ni catalyst? Is it possible that a dual mechanism exists where some steps take place at the interfaces and others in other Ni catalytic sites even if they are far from the interface? How is the oxide distributed in the Ni structure?
2. It is also concluded that the support reduces the corrosion of Ni thus extending the catalyst life. Although the stability experiments reinforce this idea, the mechanisms behind the prevention of Ni oxidation are not clear. The authors should clearly demonstrate this point.
3. How does the number of oxygen vacancies in the oxide affect the catalysis and stability?
4. In page 13 it is claimed that the introduction of Yb_2O_3 can significantly reduce the Ni particle size. But for example for a molar ratio of 90:10, how is the oxide distributed and what would be the structure of the interface? Are the two materials mixed or separated at the atomic level? Chemical link is suggested from the TEM analysis, but is there any other proof beyond visualization? The EXAFS and XPS studies give some hints but still the atomic interfacial structure is difficult to comprehend.
5. Why is 90:10 the best ratio for the catalytic activity?
6. The models used for the theoretical studies show a perfect interface between the two materials. The computational results suggest that the reaction occurs at interfaces. How about the rest of the Ni sites? This takes us to the previous question, what is the true microscopic structure of the catalyst?

Reviewer #3 (Remarks to the Author):

Sun et al. screened the HER properties of a series of Ni/ Ln_2O_3 hybrids and found an excellent HER catalyst of Ni/ Yb_2O_3 . Ni/ Yb_2O_3 exhibits a low overpotential (20 mV at 10 mA cm^{-2}) and long-term durability (360 h at 500 mA cm^{-2}). The authors claimed that Yb_2O_3 plays as a promoter to reduce the energy barrier of water dissociation, and meanwhile optimize the free energy of hydrogen adsorption and avoid the oxidation corrosion of Ni. I appreciate the heavy workload to search and optimize the catalytic activity. However, several issues still exist and listed as below.

1. The innovation of the article is not high enough, since many interface systems (oxides/Ni, sulfides/Ni, nitrides/Ni, et al.) have been reported for boosting HER performance. Moreover, the

incorporated oxides have been extensively regarded as water dissociation promoter. Therefore, this manuscript does not deliver novel design and understanding on HER catalysis.

2. For the performance, the authors claimed they achieve the best performance among the Ni-based catalysts, as shown in Figure 5f. Actually, they missed some key papers such as Ni₃N/Ni (Nat. Commun. 2018, 9, 4531) and Ni₄Mo/MoO₂ (Nat. Commun., 2017, 8, 15437), which are better than this work.

3. The fitting of Yb 4d XPS spectra in Figure 4h is strange for the peak of Yb²⁺ is unpaired and the same valence of Yb³⁺ show two pairs of peaks at different binding energies.

4. The saturated calomel electrode is unstable under alkaline conditions. It is suitable to use Hg/HgO electrode.

5. The color of the atoms in DFT calculations should be label in Figure 6. The adsorption energy of OH⁻ should be added. In addition, what's the experimental basis for the model construction of Ni(111)/Yb₂O₃(222) interface?

Response to reviewers' comments

Reviewer #1:

The paper by Sun et al. deals with the bixbyite-type Ln_2O_3 as remarkable promoters of metallic Ni for alkaline electrocatalytic hydrogen evolution. This work is a very good contribution to the field and could be published in Nature Communications after minor revision as mentioned below:

1. CAS number, purities and providers of all chemicals (including the used gases) should be added in the experimental section.

Response: Thank you for your suggestion. The CAS number, purities and providers of all chemicals (including the used gases) were added in the experimental section. *Please see Page 2, the 1st paragraph, in the revised Supplementary Information.*

2. Relative error should be added to all values, tables and figures used in the manuscript (and not some of them). This error should determine the number of digits used after the decimal point. Is it relevant for instance to use 2 numbers after the decimal point for measurement realized by EDX?

Response: Thank you for this important reminder. Relative errors were added to all available cases, which actually could determine the number of digits used after the decimal point. For example, the 2 numbers after the decimal point in EDX was realized by the experimental measurements. *Please see Figs. 2c, 2e, 2f, Fig. 3b in the revised manuscript as well as Supplementary Figs. 22, 30, 44, Supplementary Tables 1, 2 in the revised Supplementary Information.*

3. EDX should be redrawn using origin. The table should be removed from inset.

Response: EDX spectra were redrawn using origin and the tables were removed from inset. The Ni:Ln molar ratios determined by EDX were placed in the corresponding figure captions. *Please see Supplementary Figs. 7-15, 27, 41 in the revised Supplementary Information.*

Reviewer #2:

This manuscript reports a new type of catalyst where Ni nanoparticles are mixed with a small proportion of lanthanide oxides and tested for the hydrogen evolution reaction. Interestingly, it is proved that some members of the series Ln_2O_3 are very well suited for catalysts of HER. The paper is based on a series of experiments and first principles computations. The outcome is interesting and potentially very useful. I would be in favor of publication, but would like to see some clarifications of the following points.

1. It is difficult to understand what are the active sites in the mixed catalyst. From the computations it is suggested that the interfaces between Ni and the oxide is where the reaction may take place. However, why is the number of active sites enhanced for a >90% Ni catalyst? Is it possible that a dual mechanism exists where some steps take place at the interfaces and others in other Ni catalytic sites even if they are far from the interface? How is the oxide distributed in the Ni structure?

Response: We sincerely appreciate this valuable comment. To address this concern, we have taken additional experiments and DFT calculations to further clarify the active sites, catalytic mechanism and oxide distribution in the Ni structure. The details are shown below.

(i) Active sites. The active sites of catalysts were quantified by the electrochemical active surface areas (ECSAs) derived from the double-layer capacitances, which include all accessible sites in Ni/Yb₂O₃ regardless of activity. Therefore, the number of active sites is an estimated value, which includes Ni sites around the interface, Ni sites far from the interface, Yb₂O₃ sites around the interface and Yb₂O₃ sites far from the interface. As proved by the XRD patterns and TEM images, the introduction of Yb₂O₃ can lower the size of Ni nanoparticles, thereby improving the ESCA. However, when excessive Yb₂O₃ is doped, the ECSA of Ni/Yb₂O₃ is decreased, because pure Yb₂O₃ with smaller ECSA will lower the total ECSA of the Ni/Yb₂O₃ hybrid (**Supplementary Figs. 28-29**). Thus, Ni/Yb₂O₃-90:10 shows the highest ECSA among all the Ni/Yb₂O₃ hybrids (**Supplementary Fig. 30**), due to the optimum balance between the increased ECSA by reduced size of Ni nanoparticles and the depressed ECSA by doped Yb₂O₃. *Please see Pages 13-14, lines 243-250 in the revised manuscript and Supplementary Figs. 28-30 in the revised Supplementary Information.*

Supplementary Figure 28. Double-layer capacitance (C_{dl}) measurements in 1 M KOH. CV curves at different scan rates within the non-Faradaic potential range for **a** Ni/Yb₂O₃-99:1, **b** Ni/Yb₂O₃-97:3, **c** Ni/Yb₂O₃-95:5, **d** Ni/Yb₂O₃-80:20, **e** Ni/Yb₂O₃-70:30, **f** Ni/Yb₂O₃-60:40, and **g** Yb₂O₃.

Supplementary Figure 29. Double-layer capacitance (C_{dl}). Capacitive currents on the basis of scan rate for Ni, Ni/Yb₂O₃-99:1, Ni/Yb₂O₃-97:3, Ni/Yb₂O₃-95:5, Ni/Yb₂O₃-90:10 (i.e. Ni/Yb₂O₃), Ni/Yb₂O₃-80:20, Ni/Yb₂O₃-70:30, Ni/Yb₂O₃-60:40, and Yb₂O₃ electrodes.

Supplementary Figure 30. Comparison of ECSAs for Ni and Ni/Yb₂O₃ electrodes with different Ni:Yb molar ratios. The error bars represent the standard deviation based on triplicate measurements.

(ii) Catalytic mechanism. We fully agree that a dual mechanism may exist where some steps take place at the interface and others take place at the Ni sites far from the interface (*Please see Supplementary Fig. 45a in the revised Supplementary Information*). To understand the intrinsic activity of the Ni sites far from the interface, further DFT calculations were taken. The results indicate that the Ni sites far from interface perform high energy barrier of water dissociation and quite negative ΔG_{H^*} , which are almost identical to those of bare Ni, but are much lower than those of the Ni/Yb₂O₃ interfaces containing synergetic

Ni sites and Yb sites (*Supplementary Figs. 65-68*). Therefore, the alkaline HER occurs preferentially and rapidly at the interfaces of Ni/Yb₂O₃ with high intrinsic activity. *Please see Pages 27-28, lines 508-520 in the revised manuscript and Supplementary Figs. 45a, 65-68 in the revised Supplementary Information.*

Supplementary Figure 45. a Schematic illustration of alkaline HER on the heterosurface of Ni/Yb₂O₃.

Supplementary Figure 65. Atomic configurations of simulated H₂O dissociation process on Ni sites far from the interface of Ni(111)/Yb₂O₃(222). **a** Top view. **b** Side view. The blue, red, white and green spheres represent the Ni, O, H and Yb atoms, respectively.

Supplementary Figure 66. DFT calculations. **a** DFT calculated reaction energy diagram of H₂O dissociation for bare Ni, Ni sites far from the interface of Ni/Yb₂O₃ (abbreviated as Ni/Yb₂O₃-Ni site) and Ni sites far from the interface of Ni/CeO₂ (abbreviated as Ni/CeO₂-Ni site). **b** Calculated ΔG_{H^*} for bare Ni, Ni/Yb₂O₃-Ni site, and Ni/CeO₂-Ni site.

Supplementary Figure 67. Adsorption configurations of H* on the Ni sites far from the interfaces of Ni(111)/Yb₂O₃(222) and Ni(111)/CeO₂(111). **a** Top view. **b** Side view. The blue, red, white, green, and buff spheres represent the Ni, O, H, Yb, and Ce atoms, respectively.

Supplementary Figure 68. Atomic configurations of simulated H₂O dissociation process on Ni sites far from the interface of Ni(111)/CeO₂(111). **a** Top view. **b** Side view. The blue, red, white and buff spheres represent the Ni, O, H and Ce atoms, respectively.

(iii) Oxide distribution in the Ni structure. To illustrate show how Yb₂O₃ nanoparticles are distributed in the Ni structure, TEM-EDX elemental mappings and HRTEM images (*Supplementary Figs. 32-38*) of the Ni/Yb₂O₃ hybrids with different Ni:Yb molar ratios were tested and analyzed. The results show that Yb₂O₃ nanocrystallines and Ni nanoparticles are combined in the form of heterojunction structures, and Yb₂O₃ oxide nanocrystallines with smaller size are distributed around Ni nanoparticles. *Please see Page 15, lines 268-276 in the revised manuscript and Supplementary Figs. 32-38 in the revised Supplementary Information.*

Supplementary Figure 32. Characterization of Ni/Yb₂O₃-90:10. **a** TEM image (scale bar: 200 nm). **b** HRTEM (scale bar: 5 nm), where yellow and red dotted lines represent Yb₂O₃ and Ni, respectively, identified by the lattice fringes. **c** Schematic illustration of Ni/Yb₂O₃-90:10 heterostructure (black and cyan spheres represent Ni and Yb₂O₃, respectively).

Supplementary Figure 33. Characterization of Ni/Yb₂O₃-99:1. **a, b, c, d** TEM-EDX elemental mappings (scale bar: 100 nm). **e** HRTEM image (scale bar: 5 nm), where yellow and red dotted lines represent Yb₂O₃ and Ni, respectively, identified by the lattice fringes. **f** Schematic illustration of Ni/Yb₂O₃-99:1 heterostructure (black and wathet spheres represent Ni and Yb₂O₃, respectively).

Supplementary Figure 34. Characterization of Ni/Yb₂O₃-97:3. **a, b, c, d** TEM-EDX elemental mappings (scale bar: 100 nm). **e** HRTEM image (scale bar: 5 nm), where yellow and red dotted lines represent Yb₂O₃ and Ni, respectively, identified by the lattice fringes. **f** Schematic illustration of Ni/Yb₂O₃-97:3 heterostructure (black and wathet spheres represent Ni and Yb₂O₃, respectively).

Supplementary Figure 35. Characterization of Ni/Yb₂O₃-95:5. **a, b, c, d** TEM-EDX elemental mappings (scale bar: 100 nm). **e** HRTEM image (scale bar: 5 nm), yellow and red dotted lines represent Yb₂O₃ and Ni, respectively, identified by the lattice fringes. **f** Schematic illustration of Ni/Yb₂O₃-95:5 heterostructure (black and wathet spheres represent Ni and Yb₂O₃, respectively).

Supplementary Figure 36. Characterization of Ni/Yb₂O₃-80:20. **a, b, c, d** TEM-EDX elemental mappings (scale bar: 100 nm). **e** HRTEM image (scale bar: 5 nm), where yellow and red dotted lines represent Yb₂O₃ and Ni, respectively, identified by the lattice fringes. **f** Schematic illustration of Ni/Yb₂O₃-80:20 heterostructure (black and wathet spheres represent Ni and Yb₂O₃, respectively).

Supplementary Figure 37. Characterization of Ni/Yb₂O₃-70:30. **a, b, c, d** TEM-EDX elemental mappings (scale bar: 100 nm). **e** HRTEM image (scale bar: 5 nm), where yellow and red dotted lines represent Yb₂O₃ and Ni nanoparticles, respectively, identified by the lattice fringes. **f** Schematic illustration of Ni/Yb₂O₃-70:30 heterostructure (black and wathet spheres represent Ni and Yb₂O₃, respectively).

Supplementary Figure 38. Characterization of Ni/Yb₂O₃-60:40. **a, b, c, d** TEM-EDX elemental mappings (scale bar: 100 nm). **e** HRTEM image (scale bar: 5 nm), where yellow and red dotted lines represent Yb₂O₃ and Ni, respectively, identified by the lattice fringes. **f** Schematic illustration of Ni/Yb₂O₃-60:40 heterostructure (black and wathet spheres represent Ni and Yb₂O₃, respectively).

2. It is also concluded that the support reduces the corrosion of Ni thus extending the catalyst life. Although the stability experiments reinforce this idea, the mechanisms behind the prevention of Ni oxidation are not clear. The authors should clearly demonstrate this point.

Response: We are grateful to this constructive suggestion. Generally, Ni can be oxidized by oxygen dissolved in the electrolyte or migrated from the counter electrodes, leading to the loss of active sites for HER. The stability experiments and post-electrolysis characterizations confirm the higher stability of Ni in Ni/Yb₂O₃ relative to bare Ni, which is mainly attributed to the following factors. First, Yb₂O₃ is chemically stable under the pH and potential range of the HER operation. Thus, the surface closely-coupled Yb₂O₃ can serve as an excellent protection shell for Ni phase, preventing the oxygen penetration to oxidize the Ni phase and thereby retaining the highly active Ni/Yb₂O₃ heterogeneous interface for HER catalysis (*Angew. Chem. Int. Ed.* **2015**, 54, 11989-11993; *Angew. Chem. Int. Ed.* **2018**, 57, 1616-1620). The low stability of the physically mixed Ni+Yb₂O₃ sample without strongly contact interface further proves this inference (*Please see Supplementary Fig. 56*). Second, coupling with Yb₂O₃ significantly lowers the adsorption energy of O₂ on the interfacial Ni sites, as proved by additional DFT results (*Please see Supplementary Fig. 69*). The weak oxygen adsorption can availablely relieve oxygen attack and erosion on the Ni sites in Ni/Yb₂O₃ hybrid, which is the underlying mechanisms behind the prevention of Ni oxidation on the Ni/Yb₂O₃ hybrid (*Angew. Chem. Int. Ed.* **2022**, 61, e202114899). *Please see Pages 24-25, lines 443-446, 448-453; Page 28, lines 521-529 in the revised manuscript and Supplementary Figs. 56, 69 in the revised Supplementary Information.*

Supplementary Figure 56. Characterization and electrocatalytic HER performances of the Ni+Yb₂O₃ electrode. a XRD patterns. **b** Polarization curves. **c** Chronopotentiometric curves.

Supplementary Figure 69. Adsorption configurations of O_2 on the active sites of Ni(111) and Ni(111)/Yb₂O₃(222). **a** Top view. **b** Side view. The blue, red, and green spheres represent the Ni, O, and Yb atoms, respectively. The O_2 adsorption energy on Ni(111), Ni(111)/Yb₂O₃(222)-site1, and Ni(111)/Yb₂O₃(222)-site2 is -1.62 eV, -1.35 eV, and -0.98 eV, respectively.

3. How does the number of oxygen vacancies in the oxide affect the catalysis and stability?

Response: We thank the referee for this key question. To verify how does the number of oxygen vacancies in Yb₂O₃ affect the catalytic activity and stability, the Ni/Yb₂O₃ hybrids with different concentrations of oxygen vacancy were designed and synthesized in the revised work. Electrochemical test results show that these catalysts exhibit similar catalytic activity and stability, which indicate that oxygen vacancies in Yb₂O₃ have little effect on the performances of Ni/Yb₂O₃. This may be because oxygen vacancies are not the key active sites of Ni/Yb₂O₃ for alkaline HER. *Please see Page 23, lines 417-422 in the revised manuscript as well as Page 57, Supplementary Fig. 52 and the associate text in the revised Supplementary Information.*

Supplementary Figure 52. Characterization and electrocatalytic HER performances of Ni/Yb₂O₃ catalysts with different oxygen vacancy concentrations (Ni/Yb₂O₃-1.0 h, Ni/Yb₂O₃-2.0 h and Ni/Yb₂O₃-4.0 h). a XRD patterns. **b** Electron paramagnetic resonance (EPR) spectra. **c** Polarization curves. **d** Chronopotentiometric curves.

4. In page 13 it is claimed that the introduction of Yb₂O₃ can significantly reduce the Ni particle size. But for example for a molar ratio of 90:10, how is the oxide distributed and what would be the structure of the interface? Are the two materials mixed or separated at the atomic level? Chemical link is suggested from the TEM analysis, but is there any other proof beyond visualization? The EXAFS and XPS studies give some hints but still the atomic interfacial structure is difficult to comprehend.

Response: Thank you for the insightful comment. The newly taken TEM images of Ni/Yb₂O₃-90:10 in the revised work show that Yb₂O₃ nanocrystallines and Ni nanoparticles are combined in the form of hetero-junction structures. The Yb₂O₃ oxide nanocrystallines with smaller sizes are distributed on the surface of Ni nanoparticles, and clear interfaces could be observed in the heterostructures (*Please see Fig. 3d and Supplementary Fig. 32*). In the interfaces, the Ni and Yb₂O₃ nanoparticles are tightly coupled, and the Yb₂O₃ nanocrystallines epitaxially grow out of the Ni structure (*Please see Supplementary Figs. 32, 39*). Line-scanning intensity profile from the HRTEM image of Ni/Yb₂O₃ suggests Ni-O bands may exist in the

interface (*Please see Supplementary Fig. 39i*). The inverted FFT (IFFT) images and the corresponding schematic structural diagrams illustrate the phase interface of Ni and Yb₂O₃ in detail, offering a deep insight into the heterostructure (*Please see Figs. 3g, 3h*). Beyond the visual TEM images, EELS was also used to clarify the structure of the interface (*Please see Supplementary Fig. 40*). The results clearly exhibit the distribution of Ni and its interface with Yb₂O₃, and also confirm the chemical links (i.e., Ni-O bonds) between Ni and Yb₂O₃. *Please see Pages 15-17, lines 277-308 in the revised manuscript.*

Fig. 3 Structural characterizations of Ni/Yb₂O₃. **d, e** HRTEM images of Ni/Yb₂O₃ (scale bar: 5 nm for **d** and 1 nm for **e**), and line scan of HRETEM image. **f** Fast Fourier transform (FFT) pattern from **e**. **g** Inverse FFT patterns corresponding to the regions of 2 and 3 in **e**. **h** Schematic diagram of the structures corresponding to the regions of 2 and 3 in **e**. The blue, red, and green spheres represent the Ni, O, and Yb atoms, respectively.

Supplementary Figure 32. Characterization of Ni/Yb₂O₃-90:10. **a** TEM image (scale bar: 200 nm). **b** HRTEM (scale bar: 5 nm), where yellow and red dotted lines represent Yb₂O₃ and Ni, respectively, identified by the lattice fringes. **c** Schematic illustration of Ni/Yb₂O₃-90:10 heterostructure (black and white spheres represent Ni and Yb₂O₃, respectively).

Supplementary Figure 39. Characterization of Ni/Yb₂O₃. a, d, g TEM images (scale bar: 5 nm). b, c, e, f, h HRTEM images (scale bar: 1 nm). i Line intensity profile for Ni and Yb₂O₃ indicated by the blue lines in HRTEM image h.

Supplementary Figure 40. Characterization of Ni/Yb₂O₃. a Line-scan electron energy loss spectroscopy (EELS). b HRTEM image of Ni/Yb₂O₃ recorded for line-scan EELS spectrum (scale bar: 5 nm). c EELS spectra collected on Yb₂O₃ phase (I), interface (II) and Ni phase (III) of Ni/Yb₂O₃ as marked in HRTEM image b. d Magnified Ni L-edge EELS plots.

5. Why is 90:10 the best ratio for the catalytic activity?

Response: Ni/Yb₂O₃-90:10 shows the best catalytic activity, which is mainly attributed to its large ECSA, high intrinsic catalytic activity and high conductivity. As discussed in response to question 1 by this reviewer, the Ni/Yb₂O₃-90:10 sample possesses the largest ECSA resulting from its proper doping amount of Yb₂O₃. The high intrinsic catalytic activity of Ni/Yb₂O₃-90:10 is attributed to its accelerated overall HER kinetics derived from the optimal balance of Ni and Yb₂O₃ components (*Supplementary Fig. 45*), which are responsible for the H₂O dissociation step and H₂ formation step of alkaline HER, respectively. In addition, the introduction of Yb₂O₃ in metal Ni can cause a marked reduction in conductivity of the hybrids (*Supplementary Fig. 46*), thereby impeding the electron transfer for catalyzing HER. As a result, the 90:10 is the best ratio of Ni:Yb in the Ni/Yb₂O₃ hybrids. Please see Pages 19-20, lines 356-377 in the revised manuscript and Supplementary Figs. 45-46 in the revised Supplementary Information.

Supplementary Figure 45. a Schematic illustration of alkaline HER on the heterosurface of Ni/Yb₂O₃. b Schematic illustration for the balance of H₂O dissociation step and H₂ formation step obtained by regulating the Ni/Yb ratios. c Dependence of alkaline HER activity on the compositions of Ni/Yb₂O₃ hybrids.

Supplementary Figure 46. Electrical conductivity measurements. The resistance values (R) of the samples were calculated by voltmeter-ammeter method¹. The electrical conductivity was determined to be 1.65×10^3 , 3.61×10^2 , 1.22×10^2 , 1.00×10^2 , 8.55×10 , 2.88 , 3.52 , 9.71×10^{-1} and 1.46×10^{-6} S m⁻¹ for Ni, Ni/Yb₂O₃-99:1, Ni/Yb₂O₃-97:3, Ni/Yb₂O₃-95:5, Ni/Yb₂O₃-90:10, Ni/Yb₂O₃-80:20, Ni/Yb₂O₃-70:30, Ni/Yb₂O₃-60:40 and Yb₂O₃, respectively. The electrical conductivity of Ni/Yb₂O₃ hybrids decreases significantly with the increase of Yb₂O₃ doping amount due to the very low electrical conductivity of Yb₂O₃.

6. The models used for the theoretical studies show a perfect interface between the two materials. The computational results suggest that the reaction occurs at interfaces. How about the rest of the Ni sites? This takes us to the previous question, what is the true microscopic structure of the catalyst?

Response: As answered in question 1 by this reviewer, there are indeed two kinds of active sites in the Ni/Yb₂O₃ catalyst, including the Ni sites in the interface and the other Ni sites far from the interface. However, the catalytic activity of the Ni sites far from interface is very low and similar to that of pure Ni catalyst, as proved by the DFT calculations. As discussed in the response to question 4 by this reviewer, the microscopic structure of the Ni/Yb₂O₃ hybrid was clearly characterized by further experiments including HRTEM, EELS and so on.

Reviewer #3:

Sun et al. screened the HER properties of a series of Ni/Ln₂O₃ hybrids and found an excellent HER catalyst of Ni/Yb₂O₃. Ni/Yb₂O₃ exhibits a low overpotential (20 mV at 10 mA cm⁻²) and long-term durability (360

h at 500 mA cm⁻²). The authors claimed that Yb₂O₃ plays as a promoter to reduce the energy barrier of water dissociation, and meanwhile optimize the free energy of hydrogen adsorption and avoid the oxidation corrosion of Ni. I appreciate the heavy workload to search and optimize the catalytic activity. However, several issues still exist and listed as below.

1. The innovation of the article is not high enough, since many interface systems (oxides/Ni, sulfides/Ni, nitrides/Ni, et al.) have been reported for boosting HER performance. Moreover, the incorporated oxides have been extensively regarded as water dissociation promoter. Therefore, this manuscript does not deliver novel design and understanding on HER catalysis.

Response: Thanks for your comment.

(i) We fully agree that many interface systems (oxides/Ni, sulfides/Ni, nitrides/Ni, et al.) have been reported for boosting HER performance. However, the electrocatalytic performances of most of these Ni-based heterojunction hybrids are still much inferior to those of Pt catalysts and cannot meet the practical requirements in water electrolysis, i.e., driving high current densities (500 mA cm⁻²) at low overpotentials also with high stability. In terms of the overpotential at 10 mA cm⁻² and Tafel slope, Ni/Yb₂O₃ outperforms most of the Ni-based HER electrocatalysts (*Please see Supplementary Table 5*). Ni/Yb₂O₃ can also reach a high current density of 500 mA cm⁻² at an overpotential of as low as 116 mV, which is lower than that (167 mV) of Pt/C(20%). **More importantly, the Ni/Yb₂O₃ electrode keeps ultrahigh stability at a large current density of 500 mA cm⁻² over 360 h, also preceding the Pt/C(20%) catalyst and other Ni-based HER electrocatalysts**, most of which only work stably at small current densities (< 100 mA cm⁻²) for dozens of hours (*Please see Supplementary Table 5*). The high activity and durability make Ni/Yb₂O₃ one of the best HER electrocatalysts in alkaline solution, and also endow it great potentials in large-scale application for industrial electrolyzer.

(ii) We also agree with the reviewer that many metal oxides have been regarded as water dissociation promoter. However, as discussed above, the activity and stability of the reported oxides/nickel electrocatalysts cannot meet the requirements of practical industrial electrolyzers. Therefore, it is still necessary to explore new oxides coupling with nickel well to further improve its catalytic activity and stability, as different metal oxides often show diverse capacities for promoting water dissociation. Our results also demonstrate that even the lanthanide metal oxides with similar properties still exhibit quite distinct promoting effects of Ni for catalyzing HER. Fortunately, we have successfully screened out a new oxide (Yb₂O₃) that is more suitable as the water dissociation promoter for metal Ni. To the best of our knowledge, the lanthanide sesquioxides (Ln₂O₃, Ln = Sm, Eu, Gd, Dy, Ho, Er, Tm, Yb, and Lu) are the

first time to be used as the H₂O-dissociation promoter for electrocatalytic HER. This work opens up the applications of cubic bixbyite-type Ln₂O₃ in electrocatalytic yields.

(iii) The rational optimization of the proportions of two components in interface systems to synergistically promote both water dissociation and H₂ formation steps of alkaline HER has been systematically studied. This work reveals that effectively balancing the elementary steps for alkaline HER through modulating the Ni/Yb₂O₃ ratios can optimize the alkaline HER activity to the greatest extent. Moreover, the conductivity and Ni particle sizes influenced by doped Yb₂O₃ have also been precisely regulated. These results provide the useful insights on the structure-activity relationships of the interface systems for electrocatalysis.

(iv) The DFT computation results elucidate the effect of Ni/Yb₂O₃ interface on some crucial properties regulating the activity of alkaline HER, encompassing ΔG_{H^*} , adsorption energy of H₂O, energy barrier of H₂O dissociation and adsorption energy of OH⁻. The DFT results also reveal the adsorption energy of O₂ has an important effect on the HER stability of Ni catalysts. These results deliver new understanding on activity and stability of HER catalysts.

Therefore, we believe that this work deliver novel design and understanding on HER catalysis, after the full revisions according to the suggestions of all reviewers.

- 2 For the performance, the authors claimed they achieve the best performance among the Ni-based catalysts, as shown in Figure 5f. Actually, they missed some key papers such as Ni₃N/Ni (*Nat. Commun.* 2018, 9, 4531) and Ni₄Mo/MoO₂ (*Nat. Commun.*, 2017, 8, 15437), which are better than this work.

Response: Thank you very much for your kind reminding. In this manuscript, we claimed that among the Ni/Yb₂O₃ electrodes with different compositions, Ni/Yb₂O₃-90:10 shows the best performances... (*Please see Page 19, lines 354-356 in the manuscript*), and the Ni/Yb₂O₃ hybrid outperforms most of the Ni-based HER electrocatalysts (*Please see Page 23, line 424 in the manuscript*). Nevertheless, the two papers suggested by this reviewer are very important and our Figure 5f provides misleading information to readers.

The Figure 5f and Supplementary Table 5 were revised accordingly. Notably, the Ni₃N/Ni (*Nat. Commun.* 2018, 9, 4531) and Ni₄Mo/MoO₂ (*Nat. Commun.* 2017, 8, 15437) show the higher activities than Ni/Yb₂O₃ in terms of the overpotential at 10 mA cm⁻². However, Ni₃N/Ni only remains stable for a few dozen hours at a small current density of 10 mA cm⁻², and Ni₄Mo/MoO₂ exhibits decay at 200 mA cm⁻² below 30 h. In this work, the Ni/Yb₂O₃ electrode could remain stable at a large current density of ~500 mA cm⁻² for 360 h. This indicates Ni/Yb₂O₃ can meet the requirements of industrial electrolyzers that usually operate stably at high current densities above 500 mA cm⁻².

3. The fitting of Yb 4d XPS spectra in Figure 4h is strange for the peak of Yb²⁺ is unpaired and the same valence of Yb³⁺ show two pairs of peaks at different binding energies.

Response: Thank you for careful reviewing and bringing the oversight to our attention. We have carefully checked and analyzed the Yb 4d XPS spectra in Figure 4h and found that the peak of Yb²⁺ is wrong, which should be assigned to Yb³⁺. We are sorry for the mistake, which has been revised. *Please see Page 19, Figure 4h in the revised manuscript and Supplementary Figs. 4, 53 in the revised Supplementary Information.*

4. The saturated calomel electrode is unstable under alkaline conditions. It is suitable to use Hg/HgO electrode.

Response: Thanks for this very useful comment. According to this suggestion, we have further measured the HER activity and long-term stability of Ni/Yb₂O₃, Ni and Pt/C(20%) electrodes using the Hg/HgO electrode. The LSV polarization curves and long-term chronoamperometry curves of Ni/Yb₂O₃, Ni and Pt/C(20%) electrodes tested using Hg/HgO electrode almost overlap with those curves tested using saturated calomel electrode (*Supplementary Fig. 51*). *Please see Page 23, lines 415-417 in the revised manuscript and Supplementary Fig. 51 in the revised Supplementary Information.*

Supplementary Figure 51. HER performances of Ni/Yb₂O₃, Ni, Ni/CeO₂ and Pt/C(20%) tested by using different reference electrodes. a CV curves of platinum plate electrode recorded at a scan rate 5 mV s⁻¹ for potential calibration of reference electrode, with the CV result of RHE calibration $E_{(RHE)} = E_{(Hg/HgO)} + 0.921$ V. **b** LSV polarization curves. **c** Chronopotentiometric curves.

5. The color of the atoms in DFT calculations should be label in Figure 6. The adsorption energy of OH should be added. In addition, what's the experimental basis for the model construction of Ni(111)/Yb₂O₃(222) interface?

Response: Thank you for insightful suggestion. The color of the atoms in DFT calculations was labeled in Figure 6 and the other Figures for DFT calculations (*Please see Figure 6 in the revised manuscript and Supplementary Figs. 57, 58, 60-69 in the revised Supplementary Information*). The adsorption energy of OH on pure Ni, Ni/Yb₂O₃, Ni/CeO₂ and pure Yb₂O₃ were also added (*Please see the text in Page 64 in the revised Supplementary Information*). Based on the DFT analysis, the OH-binding energy on Ni/Yb₂O₃ is stronger than that on pure Ni, which is consistent with the OH adsorption/desorption experiments. For constructing the model of Ni/Yb₂O₃ hybrid, HRTEM was used to visually reveal the microscopic structure of Ni/Yb₂O₃ hybrid. As showed by the HRTEM image of Ni/Yb₂O₃, the Ni and Yb₂O₃ nanoparticles are combined in the form of heterojunction structures (*Please see Figs. 3d, 3e in the revised manuscript and Supplementary Fig. 39 in the revised Supplementary Information*). Also, it can be seen that the Ni(111) and Yb₂O₃(222) planes are the main crystal faces detected in the TEM images (*Please see Figs. 3d, 3e and Supplementary Fig. 39*), which are in good agreement with the XRD patterns of Ni and Yb₂O₃, where Ni(111) and Yb₂O₃(222) planes show the strongest diffraction peaks (*Please see Supplementary Fig. 7a*), respectively. Thus, Ni(111) and Yb₂O₃(222) were selected as the representative planes to build the model. Moreover, Ni(111) planes and Yb₂O₃(222) planes are linked in various angles, among which the parallel connection way was observed in multiple regions of the TEM images (*Please see Fig. 3e and Supplementary Figs. 39b, 39e, 39h*). Thus, we chose the parallel connection way to construct the theoretical model for the interface of Ni(111)/Yb₂O₃(222). Notably, the Ni and Yb₂O₃ particles with different sizes and planes coexist in the experimental system, and the theoretical model is only confined to a representative and optimal system. *Please see Page 25, lines 460-467 in the revised manuscript as well as Supplementary Fig. 58 and the text in Page 64 in the revised Supplementary Information.*

Fig. 3 Structural characterizations of Ni/Yb₂O₃. **d, e** HRTEM images of Ni/Yb₂O₃ (scale bar: 5 nm for **d** and 1 nm for **e**), and line scan of HRETEM image.

Supplementary Figure 39. Characterization of Ni/Yb₂O₃. **a, d, g** TEM images (scale bar: 5 nm). **b, c, e, f, h** HRTEM images (scale bar: 1 nm). **i** Line intensity profile for Ni and Yb₂O₃ indicated by the blue lines in HRTEM image **h**.

Supplementary Figure 7. Characterization of Ni/Yb₂O₃. a XRD pattern.

Supplementary Figure 58. Adsorption configurations of OH on optimized sites of different samples. a Top view for Yb₂O₃(222) and Ni(111)/Yb₂O₃(222). b Side view for Yb₂O₃(222) and Ni(111)/Yb₂O₃(222). c Top view for Ni(111) and Ni(111)/CeO₂(111). d Side view for Ni(111) and Ni(111)/CeO₂(111). The blue, red, white, green, and buff spheres represent the Ni, O, H, Yb, and Ce atoms, respectively.

Discussion on Supplementary Figure 58: Based on the results of DFT calculation, the adsorption energies of OH on the surface of Yb₂O₃(222), Ni(111)/Yb₂O₃(222)-site1 and Ni(111)/Yb₂O₃(222)-site2 are -0.24 eV, -0.12 eV and -0.54 eV, respectively. And those on the surface of Ni(111), Ni(111)/CeO₂(111)-site1 and Ni(111)/CeO₂(111)-site2 are -0.06 eV, -0.52 eV and -0.34 eV, respectively. The results reveal that the adsorption energy of OH on the surface of Ni(111) is significantly weaker than those on the surfaces of Yb₂O₃(222), Ni(111)/Yb₂O₃(222) and Ni(111)/CeO₂(111).

REVIEWERS' COMMENTS

Reviewer #2 (Remarks to the Author):

The authors have responded satisfactorily to most points.

I am in favor of recommending the paper for publication.

Regarding the answers, I have a couple of comments:

1. The newly computed interfaces shown in Figure 67 (SI) look very strained. Please make sure that the interfaces are sufficiently relaxed and minimizing the interfacial energy.
2. The proof for the anti-corrosion behavior looks oversimplified, I would expect water and dissolved oxygen in the water solution to play a major role on corrosion. Maybe the authors could add a remark on this point.

Reviewer #3 (Remarks to the Author):

The authors have well addressed my concerns. I would like to recommend acceptance without further revision.

Response to reviewers' comments

Reviewer #2:

The authors have responded satisfactorily to most points. I am in favor of recommending the paper for publication. Regarding the answers, I have a couple of comments:

1. The newly computed interfaces shown in Figure 67 (SI) look very strained. Please make sure that the interfaces are sufficiently relaxed and minimizing the interfacial energy.

Response: Thank you for your important reminder. After re-optimization, a structural model of Ni/Yb₂O₃ with minimum interfacial energy was achieved. *Please see Supplementary Figs. 65, 67 in the revised Supplementary Information.*

2. The proof for the anti-corrosion behavior looks oversimplified, I would expect water and dissolved oxygen in the water solution to play a major role on corrosion. May be the authors could add a remark on this point.

Response: Thank you for your suggestion. We fully agree that water and dissolved oxygen in the water solution are expected to play a major role on corrosion. Accordingly, we have added a remark on this point in the revised manuscript. *Please see Pages 19, lines 422-423, 425-427 in the revised manuscript.*

Reviewer #3:

The authors have well addressed my concerns. I would like to recommend acceptance without further revision.

Response: We are very grateful to your support on our work.